# Interpretability at Scale: Identifying Causal Mechanisms in Alpaca

**Zhengxuan Wu**[*]**, Atticus Geiger**[*],
**Thomas Icard**, **Christopher Potts**, and **Noah D. Goodman**

Stanford University
{wuzhengx, atticusg, icard, cgpotts, ngoodman}@stanford.edu

## Abstract

Obtaining human-interpretable explanations of large, general-purpose language models is an urgent goal for AI safety. However, it is just as important that our interpretability methods are faithful to the causal dynamics underlying model behavior and able to robustly generalize to unseen inputs. Distributed Alignment Search (DAS) [23] is a powerful gradient descent method grounded in a theory of causal abstraction that has uncovered perfect alignments between interpretable symbolic algorithms and small deep learning models fine-tuned for specific tasks. In the present paper, we scale DAS significantly by replacing the remaining brute-force search steps with learned parameters – an approach we call Boundless DAS. This enables us to efficiently search for interpretable causal structure in large language models while they follow instructions. We apply Boundless DAS to the Alpaca model (7B parameters), which, off the shelf, solves a simple numerical reasoning problem. With Boundless DAS, we discover that Alpaca does this by implementing a causal model with two interpretable boolean variables. Furthermore, we find that the alignment of neural representations with these variables is robust to changes in inputs and instructions. These findings mark a first step toward faithfully understanding the inner-workings of our ever-growing and most widely deployed language models. Our tool is extensible to larger LLMs and is released publicly at https://github.com/frankaging/align-transformers.

## 1 Introduction

Present-day large language models (LLMs) display remarkable behaviors: they appear to solve coding tasks, translate between languages, engage in open-ended dialogue, and much more. As a result, their societal impact is rapidly growing, as they make their way into products, services, and people's own daily tasks. In this context, it is vital that we move beyond behavioral evaluation to deeply explain, in human-interpretable terms, the internal processes of these models, as an initial step in auditing them for safety, trustworthiness, and pernicious social biases.

The theory of causal abstraction [5,22] provides a generic framework for representing interpretability methods that faithfully assess the degree to which a complex causal system (e.g., a neural network) implements an interpretable causal system (e.g., a symbolic algorithm). Where the answer is positive, we move closer to having guarantees about how the model will behave. However, thus far, such interpretability methods have been applied only to small models fine-tuned for specific tasks [19,20,28,7] and this is arguably not an accident: the space of alignments between the variables in the hypothesized causal model and the representations in the neural network becomes exponentially

---

[*]Equal contribution.

37th Conference on Neural Information Processing Systems (NeurIPS 2023).

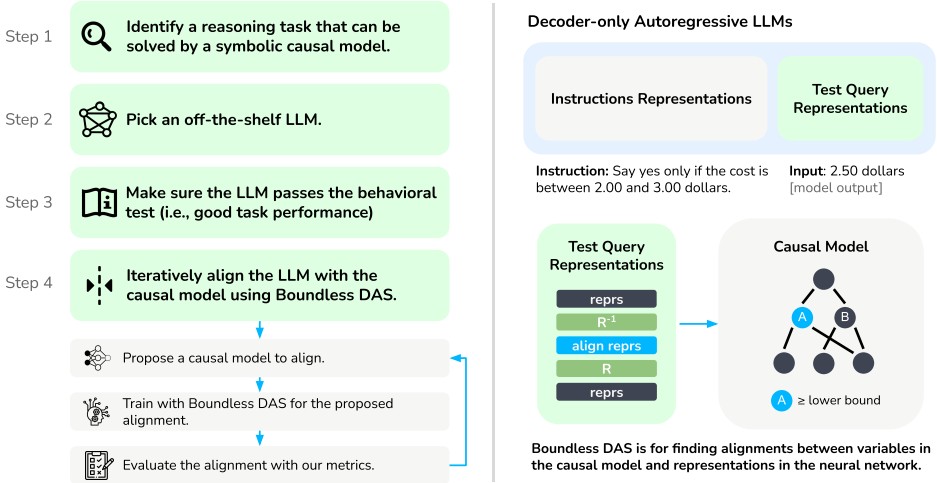

Figure 1: Our pipeline for scaling causal explainability to LLMs with billions of parameters.

larger as models increase in size. When a good alignment is found, one has specific formal guarantees. Where no alignment is found, it could easily be a failure of the alignment search algorithm.

Distributed Alignment Search (DAS) [23] marks real progress on this problem. DAS opens the door to (1) discovering structure spread across neurons and (2) using gradient descent to learn an alignment between distributed neural representations and causal variables. However, DAS still requires a brute-force search over the dimensionality of neural representations, hindering its use at scale.

In this paper, we introduce Boundless DAS, which replaces the remaining brute-force aspect of DAS with learned parameters, truly enabling interpretability at scale. We use Boundless DAS to study how Alpaca (7B) [47], an off-the-shelf instruct-tuned LLaMA model, follows basic instructions in a simple numerical reasoning task. Figure 1 summarizes the approach. We find that Alpaca achieves near-perfect task performance *because* it implements a simple algorithm with interpretable variables. In further experiments, we show that Alpaca uses this simple algorithm across a wide range of contexts and variations on the task. These findings mark a first step toward faithfully understanding the inner-workings of our largest and most widely deployed language models.

## 2 Related Work

**Interpretability** Many methods have been developed in an attempt to explain and understand deep learning models. These methods include analyzing learned weights [12,1,13], gradient-based methods [45,44,60,46,52,6], probing [14,48,24,42,12,32,8,40], syntax-driven interventions [34], self-generated model explanations [27], and training external explainers based on model behaviors [39, 31]. However, these methods rely on observational measurements of behavior and internal neural representations. Such explanations are often not guaranteed to be faithful to the underlying causal mechanisms of the target models [29,20,50,53].

**Causal Abstraction** The theory of *causal abstraction* [41,4,5] offers a unifying mathematical framework for interpretability methods aiming to uncover interpretable causal mechanisms in deep learning models [19,28,20,58,54,23] and training methods for inducing such interpretable mechanisms [21,59,58,25]. Causal abstraction [22] can represent many existing interpretablity methods, including interative nullspace projection [38,15,30], causal mediation analysis [51,33], and causal effect estimation [18,16,2]. In particular, causal abstraction grounds the research program of *mechanistic interpretability*, which aims to reverse engineer deep learning models by determining the algorithm or computation underlying their intelligent behavior [35,17,36,7,54]. To the best of our knowledge, there is no prior work that scales these methods to large, general-purpose LLMs.

**Training LLMs to Follow Instructions** Instruction-based fine-tuning of LLMs can greatly enhance their capacity to follow natural language instructions [10,37]. In parallel, this ability can also be induced into the model by fine-tuning base models with hundreds of specific tasks [56,11]. Recently,

Wang et al. [55] show that the process of creating fine-tuning data for instruction following models can partly be done by the target LLM itself ("self-instruct"). Such datasets have led to many recent successes in lightweight fine-tuning of ChatGPT-like instruction following models such as Alpaca [47], instruct-tuned LLaMA [49], StableLM [3], Vicuna [9]. Our goal is to scale methods from causal abstraction to understand *how* these models follow a particular instruction.

## 3 Methods

### 3.1 Background on Causal Abstraction

**Causal Models** We represent black box networks and interpretable algorithms using causal models that consist of *variables* taking on *values* according to *causal mechanisms*. We distinguish a set of *input variables* with possible values **Inputs**. An *intervention* is an operation that edits some causal mechanisms in a model. We denote the output of a causal model $\mathcal{M}$ provided an input $\mathbf{x}$ as $\mathcal{M}(\mathbf{x})$.

**Interchange Intervention** Begin with a model $\mathcal{M}$ (e.g., causal model or neural model) and a *base* input $\mathbf{b}$ and *source* inputs $\{\mathbf{s}_j\}_1^k$, all elements of **Inputs**, together with disjoint sets of target variables $\{\mathbf{Z}_j\}_1^k$ we are aligning. An *interchange intervention* yields a new model $\text{INTINV}(\mathcal{M}, \{\mathbf{s}_j\}_1^k, \{\mathbf{Z}_j\}_1^k)$ which is identical to $\mathcal{M}$ except the values for each set of target variables $\mathbf{Z}_j$ are fixed to be the value they would have taken for source input $\mathbf{s}_j$. We then denote the intervened output (i.e., counterfactual output) for the base input with $\text{INTINV}(\mathcal{M}, \{\mathbf{s}_j\}_1^k, \{\mathbf{Z}_j\}_1^k)(\mathbf{b})$.

**Distributed Interchange Intervention** Let $\mathbf{N}$ be a subset of variables in $\mathcal{M}$, the *target variables*. Let $\mathbf{Y}$ be a vector space with orthogonal subspaces $\{\mathbf{Y}_j\}_1^k$. Let $\mathbf{R}$ be an invertible function $\mathbf{R} : \mathbf{N} \to \mathbf{Y}$. A *distributed interchange intervention* yields a new model $\text{DII}(\mathcal{M}, \mathbf{R}, \{\mathbf{s}_j\}_1^k, \{\mathbf{Y}_j\}_1^k)$ which is identical to $\mathcal{M}$ except the causal mechanisms have been rewritten such that each subspace $\mathbf{Y}_j$ is fixed to be the value it would take for source input $\mathbf{s}_j$.[2] Specifically, the causal mechanism for $\mathbf{N}$ is,

$$F_{\mathbf{N}}^*(\mathbf{b}) = \mathbf{R}^{-1}\left( \text{Proj}_{\mathbf{Y}_0}\Big( \mathbf{R}(F_{\mathbf{N}}(\mathbf{b})) \Big) + \sum_{j=1}^{k} \text{Proj}_{\mathbf{Y}_j}\Big( \mathbf{R}(F_{\mathbf{N}}(\mathbf{s_j})) \Big) \right) \tag{1}$$

where $\mathbf{b}$ is the *base* input, $\mathbf{Y}_0 = \mathbf{Y} \setminus \bigoplus_{j=1}^k \mathbf{Y}_j$, and $\text{Proj}_{\mathbf{Y}_0}$ or $\text{Proj}_{\mathbf{Y}_j}$ represents the orthogonal projection operator of the original rotated vector into the $\mathbf{Y}_0$ or $\mathbf{Y}_j$ subspace. $F_{\mathbf{N}}(\cdot)$ represents the causal mechanisms of causal variables $\mathbf{N}$ before any intervention, whereas $F_{\mathbf{N}}^*(\cdot)$ represents the mechanisms after the distributed intervention. We then denote the intervened output (i.e., counterfactual output) for the base input as $\text{DII}(\mathcal{M}, \mathbf{R}, \{\mathbf{s}_j\}_1^k, \{\mathbf{Y}_j\}_0^k)(\mathbf{b})$.

**Causal Abstraction and Alignment** We are licensed to claim a causal model (i.e., an algorithm) $\mathcal{A}$ is a faithful interpretation of the neural network $\mathcal{N}$ if the causal mechanisms of the variables in the algorithm *abstract* the causal mechanisms of neural representations relative to a particular *alignment*. For our purposes, each variable of a high-level model is aligned with a linear subspace in the vector space formed by rotating a neural representation with an orthogonal matrix. We use $\tau$ to represent the alignment mapping between a high-level causal variable and a neural representation.

**Approximate Causal Abstraction** Interchange intervention accuracy (IIA) is a graded measure of abstraction that computes the proportion of aligned interchange interventions on the algorithm and neural network that have the same output. The IIA for an alignment of high-level variables $Z_j$ to orthogonal subspace $\mathbf{Y}_j$ between an algorithm $\mathcal{A}$ and neural network $\mathcal{N}$ is

$$\frac{1}{|\mathbf{Inputs}|^{k+1}} \sum_{\mathbf{b}, \mathbf{s}_1, \ldots, \mathbf{s}_k \in \mathbf{Inputs}} \left[ \tau\Big( \text{DII}(\mathcal{N}, \mathbf{R}, \{\mathbf{s}_j\}_1^k, \{\mathbf{Y}_j\}_1^k)(\mathbf{b}) \Big) = \right.$$
$$\left. \text{INTINV}(\mathcal{A}, \{\tau(\mathbf{s}_j)\}_1^k, \{\mathbf{Z}_j\}_1^k))(\mathbf{b}) \right] \tag{2}$$

where $\tau$ translates from low-level causal variable values to high-level neural representation values.

---

[2]Prior works focus on all-zero or mean value representation replacement [33,54] which is less general.

## 3.2 Boundless Distributed Alignment Search

Distributed alignment search (DAS) is a method for learning an alignment between interpretable causal variables of a model $\mathcal{C}$ and fixed dimensionality linear subspaces of neural representations in a network $\mathcal{N}$ using gradient descent [23] as shown in Figure 3. Specifically, an orthogonal matrix $\mathbf{R} : \mathbf{N} \to \mathbf{Y}$ is trained to maximize interchange intervention accuracy under an alignment from each variable $Z_j$ to fixed dimensionality linear subspaces $\mathbf{Y}_j$ of the rotated vector space.

Boundless DAS is our extension of DAS that learns the dimensionality of the orthogonal linear subspaces in a $d$-dimensional vector space $\mathbf{Y}$ using a method inspired by work in neural PDE [57]. Specifically, for each high-level variable $Z_j$, we introduce a learnable continuous *boundary index* parameter $b_j$ that can take on a value between 0 and $d$, where $b_0 = 0$, $b_j < b_{j+1}$, and $b_j < d$ for all $j$. The boundary mask $\mathbf{M}_j$ for the source is a vector with $d$ values between 0 and 1 where the $k$-th element of the array is defined to be

$$(\mathbf{M}_j)_k = \mathtt{sigmoid}\left(\frac{k - b_j}{\beta}\right) * \mathtt{sigmoid}\left(\frac{b_{j+1} - k}{\beta}\right) \tag{3}$$

where $\beta$ is a temperature that we anneal through training. As $\beta$ approaches 0, the masks $\mathbf{M}_j$ converge to binary-valued vectors that together encode an orthogonal decomposition of $\mathbf{Y}$.

**Weighted Distributed Interchange Intervention** Let $\mathbf{N}$ be a subset of variables in $\mathcal{M}$, the *target variables*. Let $\mathbf{Y}$ be a vector space with $d$ dimensions and let $\{\mathbf{M}_j\}_1^k$ vectors in $[0,1]^d$. Let $\mathbf{R} : \mathbf{N} \to \mathbf{Y}$ be an invertible transformation. A *weighted distributed interchange intervention* yields a new model $\textsc{SoftDII}(\mathcal{M}, \mathbf{R}, \{\mathbf{s}_j\}_1^k, \{\mathbf{M}_j\}_1^k)$ which is identical to $\mathcal{M}$ except the causal mechanisms have been rewritten such that each source input $\mathbf{s}_j$ contributes to the setting of $\mathbf{Y}$ in proportion to its mask $\mathbf{M}_j$. Specifically, the causal mechanism for $\mathbf{N}$ is set to be

$$F_{\mathbf{N}}^*(\mathbf{b}) = \mathbf{R}^{-1}\left((\mathbf{1} - \sum_{j=1}^k \mathbf{M}_j) \circ \mathbf{R}(F_{\mathbf{N}}(\mathbf{b})) + \sum_{j=1}^k \left(\mathbf{M}_j \circ \mathbf{R}(F_{\mathbf{N}}(\mathbf{s_j}))\right)\right) \tag{4}$$

where $\circ$ is element wise multiplication and $\mathbf{1}$ is a $d$ dimensional vector where each element is 1. We then denote the output for the base input as $\textsc{SoftDII}(\mathcal{M}, \mathbf{R}, \{\mathbf{s}_j\}_1^k, \{\mathbf{M}_j\}_1^k)(\mathbf{b})$.

**Boundless DAS** Given a base input $\mathbf{b}$ and source inputs $\{\mathbf{s}_j\}_1^k$, we minimize the following objective to learn a rotation matrix $\mathbf{R}^\theta$ and masks $\{\mathbf{M}_j^\theta\}_1^k$

$$\sum_{\mathbf{b}, \mathbf{s}_1, \ldots, \mathbf{s}_k \in \mathbf{Inputs}} \mathsf{CE}\left(\textsc{SoftDII}(\mathcal{N}, \mathbf{R}^\theta, \{\mathbf{s}_j\}_1^k, \{\mathbf{M}_j^\theta\}_1^k)(\mathbf{b}), \textsc{IntInv}(\mathcal{A}, \{\tau(\mathbf{s}_j)\}_1^k, \{\mathbf{Z}_j\}_1^k))(\mathbf{b})\right) \tag{5}$$

where CE is the cross entropy loss. We anneal $\beta$ throughout training and our weighted interchange interventions become more and more similar to unweighted interchange interventions. During evaluation we snap the masks to be binary-valued to create an orthogonal decomposition of $\mathbf{Y}$ where each high-level variable $Z_j$ is aligned with a linear subspace of $\mathbf{Y}$ picked out by the mask $\mathbf{M}_j$, with the residual (unaligned) subspace being picked out by the mask $(\mathbf{1} - \sum_{j=1}^k \mathbf{M}_j)$.

**Time Complexity Analysis** DAS learns a rotation matrix but requires a manual search to determine how many neurons are needed to represent the aligning causal variable. Boundless DAS automatically learns boundaries (i.e., how many neurons are needed is determined by the "soft" boundary index via a boundary mask learning). For instance, given a representation with a dimension of 1024, DAS should in principle be run for all lengths k from 1 to 1024. In practice, this would be infeasible, so some subset of the lengths need to be chosen heuristically, which risks missing genuine structure. For Boundless DAS, we turn this search process into a mask learning process. DAS [23] is $O(n \times m)$ where $n$ is the number of total dimensions of the neural representation we are aligning and $m$ is the number of causal variables, while Boundless DAS is $O(m)$.

## 4 Experiment

### 4.1 Price Tagging

We follow the approach in Figure 1 by first assessing the ability of Alpaca to execute specific actions based on the instructions provided in the input. Formally, the input to the model $\mathcal{M}$ is given an

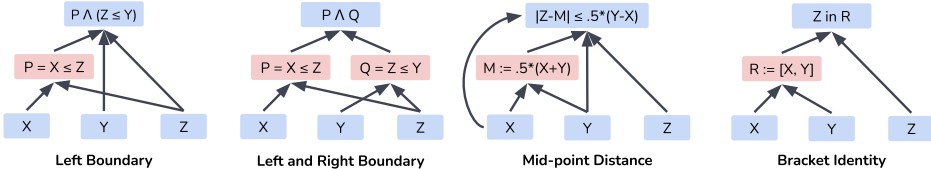

Figure 2: Four proposed high-level causal models for how Alpaca solves the price tagging task. Intermediate variables are in red. All these models perfectly solve the task.

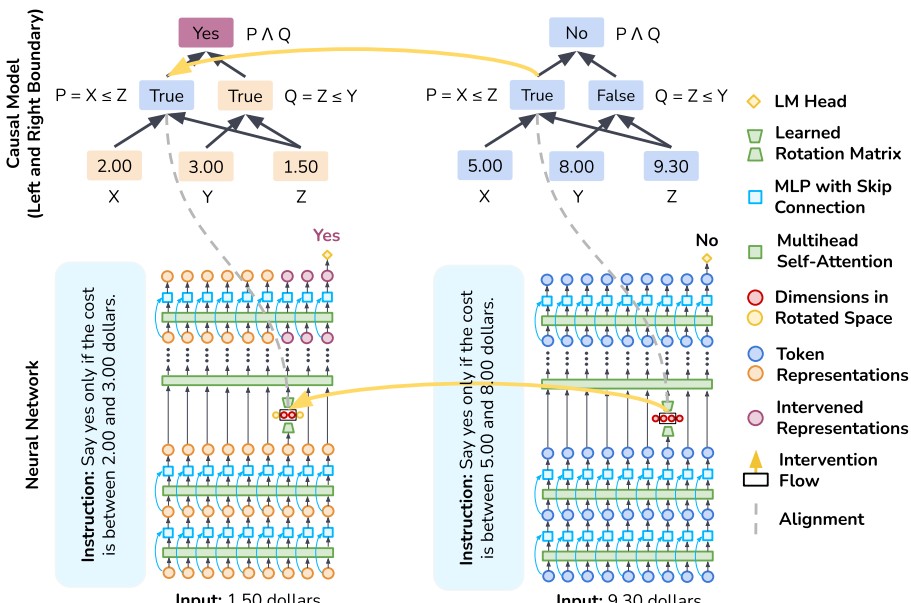

Figure 3: Aligned distributed interchange interventions performed on the Alpaca model that is instructed to solve our Price Tagging game. To train Boundless DAS, we sample two training examples and then swap the intermediate boolean values between them to produce a counterfactual output using our causal model. In parallel, we swap the aligned dimensions of the neural representations in rotated space. Lastly, we update our rotation matrix such that our neural network has a more similar counterfactual behavior to the causal model.

instruction $t_i$ (e.g., "correct the spelling of the word:") followed by a test query input $x_i$ (e.g., "aplpe"). We use $\mathcal{M}(t_i, x_i)$ to depict the model generation $y_p$ given the instruction and the test query input. We can evaluate model performance by comparing $y_p$ with the gold label $y$. We focus on tasks with high model performance, to ensure that we have a known behavioral pattern to explain.

The instruction prompt of the Price Tagging game follows the publicly released template of the Alpaca (7B) model. The core instruction contains an English sentence:

*Please say yes only if it costs between* `[X.XX]` *and* `[X.XX]` *dollars, otherwise no.*

followed by an input dollar amount `[X.XX]`, where `[X.XX]` are random continuous real numbers drawn with a uniform distribution from [0.00, 9.99]. The output is a single token 'Yes' or 'No'.

For instance, if the core instruction says *Please say yes only if it costs between* `[1.30]` *and* `[8.55]` *dollars, otherwise no.*, the answer would be "Yes" if the input amount is "3.50 dollars" and "No" if the input is "9.50 dollars". We restrict the absolute difference between the lower bound and the upper bound to be [2.50, 7.50] due to model errors outside these values – again we need behavior to explain.

## 4.2 Hypothesized Causal Models

As shown in Figure 2, we have identified a set of human-interpretable high-level causal models, with alignable intermediate causal variables, that would solve this task with 100% performance:

- **Left Boundary**: This model has one high-level boolean variable representing whether the input amount is higher than the lower bound, and an output node incorporating whether the input amount is also lower than the high bound.

- **Left and Right Boundary**: The previous model is sub-optimal in only abstracting one of the boundaries. In this model, we have two high-level boolean variables representing whether the input amount is higher than the lower bound and lower than the higher bound, respectively. We take a conjunction of these boolean variables to predict the output.

- **Mid-point Distance**: We calculate the mid-point of the lower and upper bounds (e.g., the mid-point of "3.50" and "8.50" is "6.00), and then we take the absolute distance between the input dollar amount and the mid-point as $a$. We then calculate one-half of the bounding bracket length (e.g., the bracket length for "3.50" and "8.50" is "5.00") as $b$. We predict output "Yes" if $a \leq b$, otherwise "No". We align only with the mid-point variable.

- **Bracket Identity**: This model represents the lower and upper bound in a single interval variable and passes this information to the output node. We predict the output as "Yes" if the input amount within the interval, otherwise "No".

**Model Architecture** Our target model is the Alpaca (7B) [47], an off-the-shelf instruct-tuned LLaMA model. It is a Transformer-based decoder-only autoregressive trained language model with 32 layers and 32 attention heads. It has a hidden dimension in size of 4096, which is also the dimension of our rotation matrix which is applied for each token representation. In total, the rotation matrix contains 16.8M parameters and has size $4096 \times 4096$.

**Alignment Process** We train Boundless DAS with test query token representations (i.e., starting from the token of the first input digit till the last token in the prompt) in a selected set of 7 layers: $\{0, 5, 10, 15, 20, 25, 30\}$. We also train Boundless DAS on the token before the first digit as a control condition where nothing should be expected to be aligned. We interchange with a single source, while allowing multiple causal variables to be aligned across examples. For simplicity, we enforce the interval between two boundary indices to be the same (i.e., the same size of linear subspace for each aligning causal variable). We run each experiment with three distinct random seeds. Since the global optimum of the Boundless DAS objective corresponds to the best attainable alignment, but SGD may be trapped by local optima, we report the best-performing seed in terms of IIA. Figure 3 provides an overview of these analyses.

**Evaluation Metric** To evaluate models, we use Interchange Intervention Accuracy (IIA) as defined in Eqn. 2. IIA is bounded between 0.0 and 1.0. We measure the baseline IIA by replacing the learned rotation matrix with a random one. The lower bound of IIA for "Left Boundary" and "Left and Right Boundary" is about 0.50, and for "Mid-point Distance" or "Bracket Identity" it is about 0.60. The latter two baseline IIAs are higher than chance because they are conditioned on the distribution of our output labels (i.e., how many times the original label gets to be flipped due to the intervention). See Section 4.7 for more discussion of metric calibration. Additionally, IIA can occasionally go above the model's task performance, when the interchange interventions put the model in a better state, but IIA is constrained by task accuracy for the most part.

### 4.3 Boundless DAS Results

Figure 4 shows our main results, given in terms of IIA across our four hypothesized causal models (Figure 2). The results show very clearly that 'Left Boundary' and 'Left and Right Right Boundary' (top panels) are highly accurate hypotheses about how Alpaca solves the task. For them, IIA is at or above task performance (0.85), and intermediate variable representations are localized in systematically arranged positions. By contrast, 'Mid-point Distance' and 'Bracket Identity' (bottom panels) are inaccurate hypotheses about Alpaca's processing, with IIA peaking at around 0.72.

These findings suggest that, when solving our reasoning task, Alpaca internally follows our first two high-level models by representing causal variables that align with boundary checks for both left and right boundaries. Interestingly, heatmaps on the top two rows also show a pattern of higher scores around the bottom left and upper right and close to zero scores in other positions. This is significantly different from the other two alignments where, although some positions are highlighted (e.g., the representations for the last token), all the positions receive non-zero scores. In short, the accurate hypotheses correspond to highly structured IIA patterns, and the inaccurate ones do not.

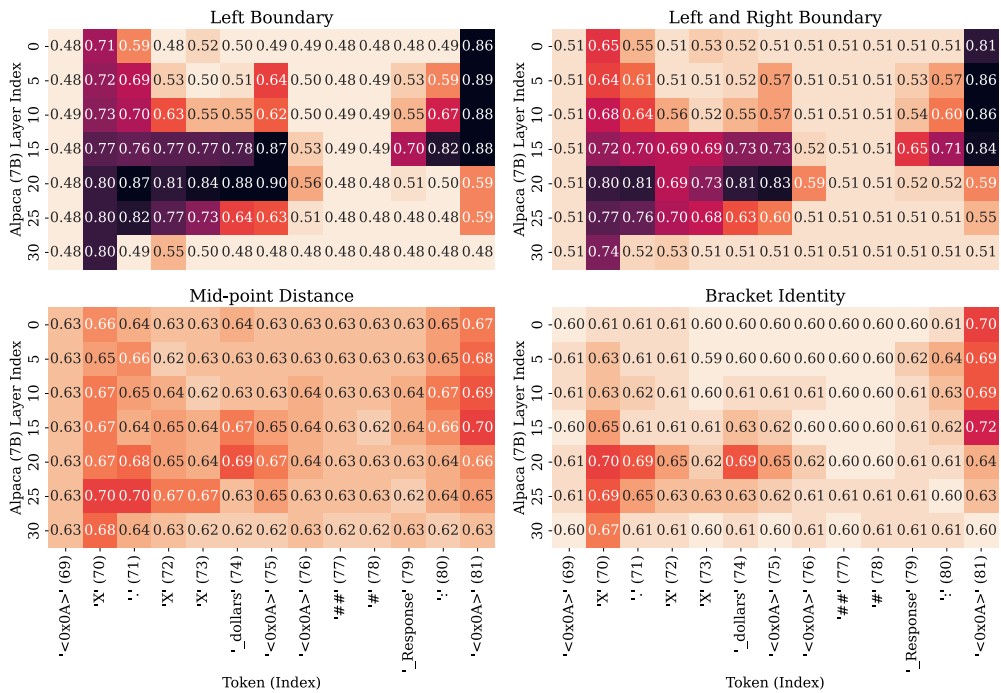

Figure 4: Interchange Intervention Accuracy (IIA) for four different alignment proposals. The Alpaca model achieves 85% task accuracy. The higher the number is, the more faithful the alignment is. We color each cell by scaling IIA using the model's task performance as the upper bound and a dummy classifier (predicting the most frequent label) as the lower bound. These results indicate that the top two are highly accurate hypotheses about how Alpaca solves the task, whereas the bottom two are inaccurate in this sense. Analyzing tokens includes special tokens (e.g., '<0x0A>' for linebreaks) required by Alpaca's instruct-tuning template as provided in Appendix A.1. Heatmap color uses min-max standardized IIA using the random rotation baseline as the lower bound, and task performance as the upper bound.

Additionally, alignments are better when the model post-processes the query input with 1–2 additional steps: accuracy is higher at position 75 compared to all previous positions. Surprisingly, there exist *bridging tokens* (positions 76–79) where accuracy suddenly drops compared to earlier tokens, which suggests that these representations have weaker causal effects on the model output. In other words, boundary-check variables are fully extracted around position 75, level 10, and are later copied into activations for the final token before responding. By comparing the heatmaps on the top row, our findings suggest that aligning multiple variables at the same time poses a harder alignment process, in that it lowers scores slightly across multiple positions.

## 4.4 Interchange Interventions with (In-)Correct Inputs

IIA is highly constrained by task performance, and thus we expect it to be much lower for inputs that the model gets wrong. To verify this, we constructed an evaluation set containing 1K examples that the model gets wrong and evaluated our 'Left and Right Boundary' hypothesis on this subset. Table 1 reports these results in terms of max IIA and the correlation of the IIA values with those obtained in our main experiments. As expected, IIA drops significantly. However, two things stand out: IIA is far above task performance (which is 0.0 by design now), and the correlation with the original IIA map remains high. These findings suggest that the model is using the same internal mechanisms to process these cases, and so it seems possible the model is narrowly missing the correct output predictions.

We also expect IIA to be higher if we focus only on cases that the model gets correct. Table 1 confirms this expectation using 'Left and Right Boundary' as representative. $IIA_{max}$ is improved from 0.86 to 0.88, and the correlation with the main results is essentially perfect. Full heatmaps for these analyses are given in Appendix A.4.

| Experiment | Task Acc. | $IIA_{max}$ | Correlation |
|---|---|---|---|
| Left Boundary (♣) | 0.85 | 0.90 | 1.00 |
| Left and Right Boundary (♥) | 0.85 | 0.86 | 1.00 |
| Mid-point Distance | 0.85 | 0.70 | 1.00 |
| Bracket Identity | 0.85 | 0.72 | 1.00 |
| Correct Only | $1.00^{†}$ | 0.88 | 0.99 (♥) |
| Incorrect Only | $0.00^{†}$ | 0.71 | 0.84 (♥) |
| New Bracket (Seen) | 0.94 | 0.94 | 0.97 (♣) |
| New Bracket (Unseen) | 0.95 | 0.95 | 0.94 (♣) |
| Irrelevant Contexts | 0.84 | 0.83 | 0.99 (♥) |
| Sibling Instructions | 0.84 | 0.83 | 0.87 (♥) |
| + *exclude* top right | 0.84 | 0.83 | 0.92 (♥) |

Table 1: Summary results for all experiments with task performance as accuracy (range $[0, 1]$), maximal interchange intervention accuracy (IIA) (range $[0, 1]$) across all positions and layers, Pearson correlations of IIA between two distributions (compared to ♣ or ♥; range $[-1, 1]$). $^{†}$This is empirical task performance on the evaluation dataset for this experiment.

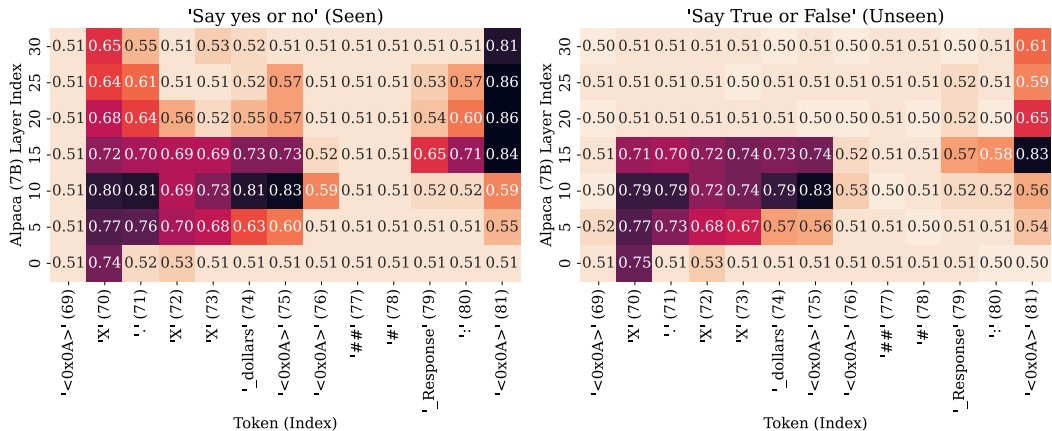

Figure 5: Interchange Intervention Accuracy (IIA) evaluated with different output formats. The seen setting is for training, and the unseen setting is for evaluation.

## 4.5 Do Alignments Robustly Generalize to Unseen Instructions and Inputs?

One might worry that our positive results are highly dependent on the specific input–output pairs we have chosen. We now seek to address this concern by asking whether the causal roles (i.e., alignments) found using Boundless DAS in one setting are preserved in new settings. This is crucial, as it tells how robustly the causal model is realized in the neural network.

**Generalizing Across Two Different Instructions**  Here, we assess whether the learned alignments for the 'Left Boundary' causal model transfer between different specific price brackets in the instruction. To do this, we retrain Boundless DAS for the high-level model with a fixed instruction that says "between 5.49 dollars and 8.49 dollars". Then, we fix the learned rotation matrix and evaluate with another instruction that says "between 2.51 dollars and 5.51 dollars". For both, Alpaca is successful at the task, with around 94% accuracy. Our hypothesis is that if the found alignment of the high-level variable is robust, it should transfer between these two settings, as the aligning variable is a boolean-type variable which is potentially agnostic to the specific comparison price.

Table 1 gives our findings in the 'New Bracket' rows. Boundless DAS is able to find a good alignment for the training bracket with an $IIA_{max}$ that is about the same as the task performance at 94%. For our unseen bracket, the alignments also hold up extremely well, with no drop in $IIA_{max}$. For both cases, the found alignments also highly correlate with the counterpart of our main experiment.

**Generalizing with Inserted Context**  Recent work has shown that language models are sensitive to irrelevant context [43], which suggests that found alignments may overfit to a set of fixed contexts.

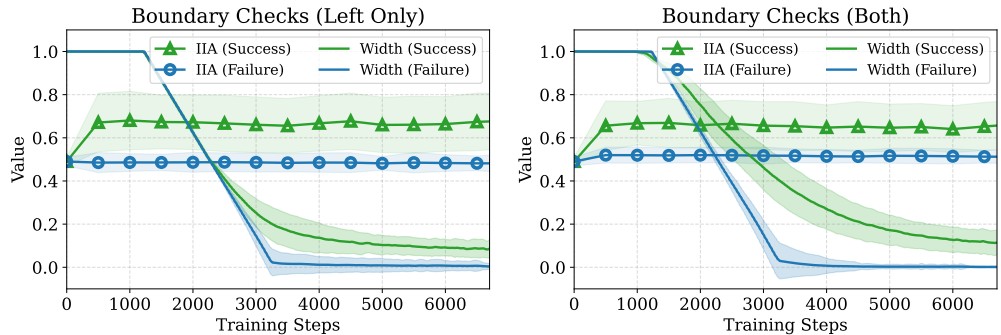

Figure 6: Learned boundary width for intervention site and in-training evaluation interchange intervention accuracy (IIA) for two groups of data: (1) aligned group where the boundary does not shrink to 0 at the end of the training; (2) unaligned group where the boundary does shrink to 0 at the end of the training. 1 on the y-axis means either 100% accuracy for IIA, or the variable is occupying half of the hidden representation for the boundary width.

To address this concern, we add prefix strings to the input instructions and evaluate how 'Left and Right Boundary' alignment transfers. We thus generate 20 random prefixes using GPT-4.[3] All of our prefixes, and our method for generating them, can be found in Appendix A.6. The model achieves 84% task performance with random contexts added as prefixes, which is slightly lower than the average task performance. Nonetheless, the results in Table 1 suggest the found alignments transfer surprisingly well here, with only 2% drop in $IIA_{max}$. Meanwhile, the mean IIA distribution across positions and layers is highly correlated with our base experiment. Thus, our method seems to identify causal structure that is robust to changes in irrelevant task details *and* position in the input string.

**Generalizing to Modified Outputs** We further test whether the alignments found for instructions with a template "Say yes . . . , otherwise no" can generalize to a new instruction with a template "Say True . . . , otherwise False". If there are indeed latent representations of our aligning causal variables, the found alignments should persist and should be agnostic to the output format.

Table 1 shows that this holds: the learned alignments actually transfer across these two sibling instructions, with a minimum 1% drop in $IIA_{max}$. In addition, the correlation increases 6% when excluding the top right corner (top 3 layers in the last position), as seen in the final row of the table. This result is further substantiated in Figure 5, where we see that IIA drops near the top right. This indicates a late fusion of output classes and working representations, leading to different computations for the new instruction only close to the output.

## 4.6 Boundary Learning Dynamics

Boundless DAS automatically learns the intervention site boundaries (Section 3.2). It is important to confirm that we do not give up alignment accuracy by optimizing boundaries, and to explore the dimensionality actually needed to represent the abstract variables. We sample 100 experiment runs from our experiment pool and create two groups: (1) the success group where the boundary does not shrink to 0 at the end of the training; (2) the failure group where the boundary does shrink to 0 at the end of the training. The failure group is for those instances where there is no significant causal role of representations found by our method. We plot how boundary width and in-training evaluation IIA vary throughout training. Figure 6 shows that, in the success cases, each aligned variable only needs 5–10% of the representation space. IIA performance converges quickly for success cases and, crucially, maintains stable performance despite shrinking dimensionality. This suggests that only a small fraction of the whole representation space is needed for aligning a causal variable, and that these representations can be efficiently identified by Boundless DAS.

---

[3]We generate these prefixes using the public chat interface `https://chat.openai.com/`.

### 4.7 Metric Calibration

One concern brought up by our reviewers is whether our metric, interchange intervention accuracy (IIA), is well calibrated. Here, we provide additional evidence. We report IIAs in the same way as in Figure 4 but with random rotation matrices. With a random rotation matrix, we find that the IIA score drops from 0.83 to 0.53 at layer 10, token position 75, for our "left and right boundary" causal model. Other positions drop significantly as well. These results calibrate IIAs in case of unbalanced labels (e.g., our two control causal models reach about 0.60 IIA at the same location with a random rotation matrix). This shows that Boundless DAS finds substantial structure in Alpaca as compared to what these random baselines can find. Details can be found in Figure 12 in the Appendix.

We can also help calibrate our result by comparing Boundless DAS applied to Alpaca with a randomly initialized LLaMA-7B. This can help quantify the extent to which DAS leverage random causal structure in order to overfit to its training objective. This random model results in near 0% task performance on our task, as expected. After running Boundless DAS on the first layer of this random LLaMA-7B model, the found alignments to each token's representation ranged from 0% to 69% IIA, which is comparable to a most frequent label dummy model (66%). For the representations with near chance IIA, this means that we were able to find a distributed neural representation that shifts the probability mass to "yes" or "no", but does not differentiate between the two. We also found that we could find a distributed neural representation that shifts the probability mass to the tokens "dog" and "give" instead of "yes" and "no". More importantly, we found that IIA drops to 0% for this run on all of our robustness checks from the paper. In addition, it drops to 0% if we use it on another randomly initialized LLaMA-7B model with a different random seed. Overall, it seems that the causal structure we did find has arisen by chance, and we might expect very large models to be more prone to such occurrences. Our robustness checks definitively clarify the situation.

## 5 Analytic Strengths and Limitations

Explanation methods for AI should be judged by the degree to which they can, in principle, identify models' true internal causal mechanisms. If we reach 100% IIA for a high-level model using Boundless DAS we can assert that we have positively identified a correct causal structure in the model (though perhaps not the only correct abstraction). This follows directly from the formal results of Geiger et al. [22]. On the other hand, failure to find causal structure with Boundless DAS is not a proof that such structure is missing, for two reasons. First, Boundless DAS explores a very flexible and large hypothesis space, but it is not completely exhaustive. For instance, a set of variables represented in a highly non-linear way might be missed. Second, we are limited by the space of causal models we think to test ourselves.

For models where task accuracy is below 100%, it is still possible for IIA to reach 100% if the high-level causal model *explains errors* of the low-level model. However, in cases like those studied here, where our high-level model matches the idealized task but our language model does not completely do so, we expect Boundless DAS to find only partial support for causal structure even if we have identified the ideal set of hypotheses and searched optimally. This too follows from the results of Geiger et al. [22]. However, as we have seen in the results above, Boundless DAS can find structure even where the model's task performance is low, since task performance can be shaped by factors like a suboptimal generation method or the rigidity of the assessment metric. Future work must tighten this connection by modeling errors in the language model in more detail.

## 6 Conclusion

We introduce Boundless DAS, a novel and effective method for scaling causal analysis of LLMs to billions of parameters. Using Boundless DAS, we find that Alpaca, off-the-shelf, solves a simple numerical reasoning problem in a human-interpretable way. Additionally, we address one of the main concerns around interpretability tools developed for LLMs – whether found alignments generalize across different settings. We rigorously study this by evaluating found alignments under several changes to inputs and instructions. Our findings indicate robust and interpretable algorithmic structure. Our framework is generic for any LLMs and is released to the public. We hope that this marks a step forward in terms of understanding the internal causal mechanisms behind the massive LLMs that are at the center of so much work in AI.

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

# A  Appendix

## A.1  Price Tagging Game Experiment

Figure 7: Instruction template for the Price Tagging game. We have three variables inside the prompt: lower bound amount, upper bound amount, and the input amount where each number has two decimal points but bounded with [0.00, 9.99].

Figure 7 shows our templates for our Price Tagging game Experiment. This format is enforced by the instruction tuning template of the Alpaca model.

## A.2  Training Details

**Training Setups**  During training for Boundless DAS, model weights are frozen except for the rotation matrix. To train the rotation matrix, we use `Adam` [26] as our optimizer with an effective batch size of 64 and a learning rate of $10^{-3}$. We use a different learning rate $10^{-2}$ for our boundary indices for quicker convergence. We train our rotation matrix for 3 epochs and evaluate with a separate evaluation set for each 200 training steps, and we save the best rotation matrix evaluated on the evaluation set during training and test on another held-out testing set. Our training set has 20K examples. Our in-training evaluation is limited to 200 examples for quicker training time, and the hold-out testing dataset has 1K examples. These datasets are generated on the fly for different causal models (i.e., different random seeds result in different training data). The rotation matrix is trained by using the orthogonalized parameterization provided by the `torch` library.[4] For boundary learning, we anneal temperature from 50.0 to 0.10 with a step number equal to the total training steps and temperature steps with gradient backpropagation. Each alignment experiment is enabled with `bfloat16` and can fit within a single A100 GPU. Each alignment experiment takes about 1.5 hrs. To finish all of our experiments, it takes roughly 2 weeks of run-time with 2 × A100 nodes with 8 GPUs each. We run experiments with 3 distinct random seeds and report the best result using IIA as our metrics.

**Vanilla Task Performance Remarks**  To ensure the model we are aligning behaviorally achieves good task performance, we evaluate the model task performance with 1K sampled triples of values more than 300 times. The Alpaca (7B) model achieves 85% for randomly drawn triples of numbers on average. We want to emphasize that it is really hard to find a good reasoning task with intermediate steps (e.g., tasks like "Return yes of it costs more than 2.50 dollars" do not require an intermediate step) that presently available 7B–30B LLMs can solve. We tested different reasoning tasks (e.g., math, logic, natural QA, code, etc.) with instruct-tuned LLaMA [49], Dolly,[5] StableLM [3] Vicuna [9], Instruct-tuned GPT-J,[6] and mostly struggled to find tasks they could do well.

---

[4]pytorch.org/docs/stable/generated/torch.nn.utils.parametrizations.orthogonal.html

[5]huggingface.co/databricks/dolly-v2-12b

[6]We use the publicly available model at `https://huggingface.co/nlpcloud/instruct-gpt-j-fp16` which is GPT-J finetuned on Alpaca.

### A.3 Counterfactual Dataset Generation

To train Boundless DAS, we need to generate counterfactual datasets where each example contains a pair of inputs as base and source, an intervention indicator (i.e., what variable we are intervening), and a counterfactual label. For instance, if we are training Boundless DAS for our Boundary Check (Left Only) high-level causal model, we may sample the base example with the bracket as [2.50, 7.50] and the input amount of 1.50 dollars, and the source example with the bracket as [3.50, 8.50] and the input amount of 9.50 dollars. If we intervene on the variable whether the input variable is higher than the lower bound, and we swap the variable from the second example into the first example, we will have a counterfactual label as "Yes". For each high-level causal model, we follow the same steps to construct the counterfactual dataset.

### A.4 Interchange Intervention Accuracy

Figure 8 to Figure **??** show a breakdown of the results presented in Table 1.

### A.5 Pseudocode for Boundless DAS

Boundless DAS is a generic framework and can be implemented for models with different architectures. Here we provide a pseudocode snippet for a decoder-only model.

```
1 def sigmoid_boundary_mask(population, boundary_x, boundary_y, temperature):
2     return torch.sigmoid((population - boundary_x) / temperature) * \
3         torch.sigmoid((boundary_y - population) / temperature)
4
5 class AlignableDecoderLLM(DecoderLLM):
6     # overriding existing forward function
7     def forward(hidden_states, source_hidden_states):
8         for idx, decoder_layer in enumerate(self.layers):
9             if idx == self.aligning_layer_idx:
10                # select aligning token reprs
11                aligning_hidden_states = hidden_states[:,start_idx:end_idx]
12                # rotate
13                rotated_hidden_states = rotate(aligning_hidden_states)
14                # interchange intervention with learned boundaries
15                boundary_mask = sigmoid_boundary_mask(
16                    torch.arange(0, self.hidden_size),
17                    self.learned_boundaries,
18                    self.temperature
19                )
20                rotated_hidden_states = (1. - boundary_mask) * rotated_hidden_states +
21                    boundary_mask * source_hidden_states
22                # unrotate
23                hidden_states[:,start_idx:end_idx] = unrotate(rotated_hidden_states)
24            hidden_states = self.layers[idx](hidden_states)
25        return hidden_states
```

A simplified version of Boundless DAS with pseudocode for a generic decoder-only model.

### A.6 Generated Irrelevant Contexts

To generate random contexts in Table 2, we prompt GPT-4@Aug05-2023 to generate short English sentences. These sentences are added to the original prompt as prefixes.

Table 2: Our prompt and **twenty** different contexts generated by `GPT4-Aug05-2023` as prefixes.

| | | |
|---|---|---|
| **Our Prompt:** I will generate random short sentences (they do not have to be in English), ranging from 3 words to 15 words. Here are two examples in English: "Pricing tag game!" and "Fruitarian Frogs May Be Doing Flowers a Favor." Generate 100 examples for me and return them in a .txt file. | | |
| We are so back! | Kiwis fly in their dreams. | Frogs leap into detective work. |
| Jumping jacks on Jupiter? | Beans in my boots? | Clouds have a fluffy sense of humor. |
| Koalas start each day with a long nap. | Dinosaurs loved a good bedtime story. | Butterflies throw colorful surprise parties. |
| Rhinos are experts at playing hide and seek. | Badgers brew the finest coffee. | Peanuts have their own underground society. |
| Pigeons love rooftop cinema nights. | Fish run deep-sea detective agencies. | Cheetahs host slow-motion replay parties. |
| Hamsters are wheel design experts. | Doves write the world's best love songs. | Porcupines are actually very cuddly. |
| Fireflies choreograph nightly light shows. | Ants host underground food festivals. | |

## A.7 Common Questions

In this section, we answer common questions that may be raised while reading this report.

*Can Boundless DAS work with 175B models?*

Yes, it does. Taking the recently released Bloom-176B as an example, the hidden dimension is 14,336, which results in a rotation matrix containing 205M parameters. This is possible if the model is sharded and the rotation matrix is put into a single GPU for training. This is only possible given the fact that Boundless DAS is now implemented as a token-level alignment tool. If the alignment is for a whole layer (i.e., token representations are concatenated), it is currently impossible. It is worth noting that training the rotation matrix consumes more resources than training a simple weight matrix, as the orthogonalization process in `torch` requires matrix exponential, which means saving multiple copies of the matrix.

*The current paper limits to a set of tasks from the same family. How well will the findings in the paper generalize to other scenarios?*

Existing public models in 7B–30B range are still not effective in solving reasoning puzzles that require multiple steps. We test different reasoning tasks as mentioned in Section A.2. Once larger LLMs are released and evaluated as having stronger reasoning abilities, Boundless DAS will be ready as an analysis tool. Although our Price Tagging game task is simple, we test with different variants of the task and show the efficacy of our method.

*Is it possible to apply Boundless DAS over the whole layer representation?*

It is possible for smaller models if we are learning a full-rank square rotation matrix. It is intractable for Alpaca at present. Let's say we align 10 token representations together. We will have a total dimension size of 40,960. This leads to a rotation matrix containing 1.6B parameters. Given the current orthogonalization process in `torch`, this will require a GPU with memory larger than the current A100 GPU just to fit in a trainable rotation matrix.

*Only limited high-level models are tested with Boundless DAS in the experiments.*

There is a large number of high-level models to solve the task, and we only test a portion of them. We think that this is sufficient to demonstrate the advantages of our pipeline. It would be interesting if humans can be in the loop of testing different hypotheses of the high-level causal model in future work. On the other hand, not all the high-level models are interesting to analyze. For instance, if two high-level models are equivalent (e.g., swapping variable names), it is not in our interest to find alignments.

*Can Boundless DAS be applied to find circuits in a model?*

Yes, it can. We believe that "circuit analysis" can be theoretically grounded in causal abstraction and is deeply compatible with our work. Previous work in this mode usually relies on the assumption of localist representation (i.e., there exists a one-to-one mapping between a group of neurons and a high-level concept). This is often too idealized, as multiple concepts can easily be aligned with the same group of neurons, or a group of neurons may map to multiple concepts. Boundless DAS actually drops this assumption by specifically focusing on distributed alignments. More importantly, Boundless DAS can be easily used in a head-wise alignment search where we add a shared rotation matrix on top of head representations of all the tokens.

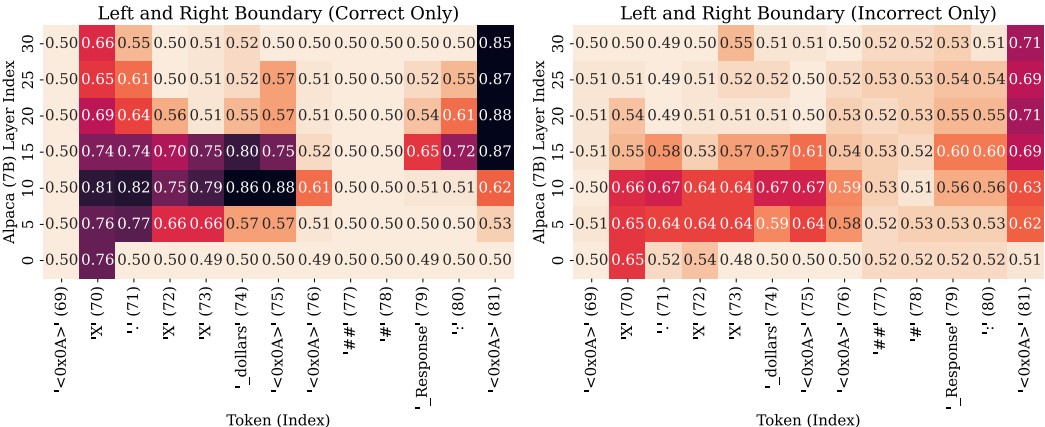

Figure 8: Interchange Intervention Accuracy (IIA) evaluated with correct input examples only as well as incorrect input examples only for our "Left and Right Boundary" causal model. The higher the number is, the more faithful the alignment is. We color each cell by scaling IIA using the model's task performance as the upper bound and a dummy classifier (predicting the most frequent label) as the lower bound.

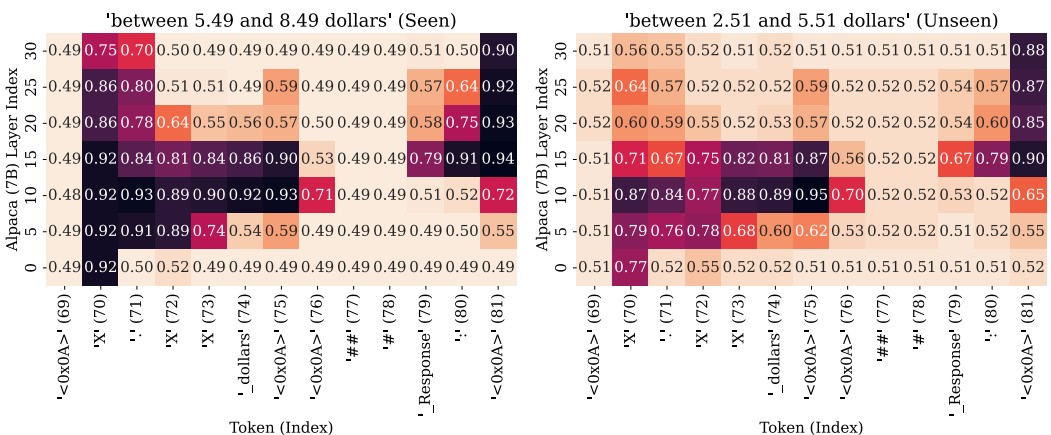

Figure 9: Interchange Intervention Accuracy (IIA) evaluated with different settings of brackets in the instruction. The seen setting is for training, and the unseen setting is for evaluation. The higher the number is, the more faithful the alignment is. We color each cell by scaling IIA using the model's task performance as the upper bound and a dummy classifier (predicting the most frequent label) as the lower bound.

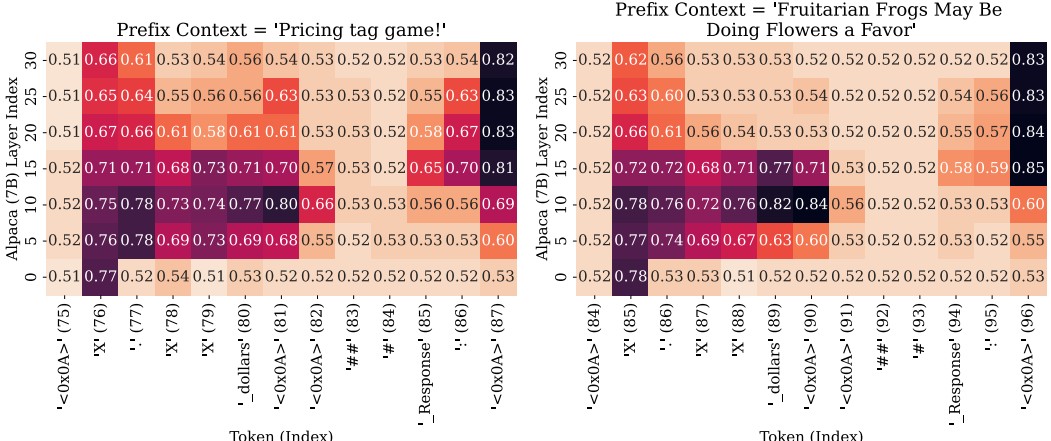

Figure 10: Interchange Intervention Accuracy (IIA) evaluated with two different irrelevant contexts inserted as prefixes. The higher the number is, the more faithful the alignment is. We color each cell by scaling IIA using the model's task performance as the upper bound and a dummy classifier (predicting the most frequent label) as the lower bound.

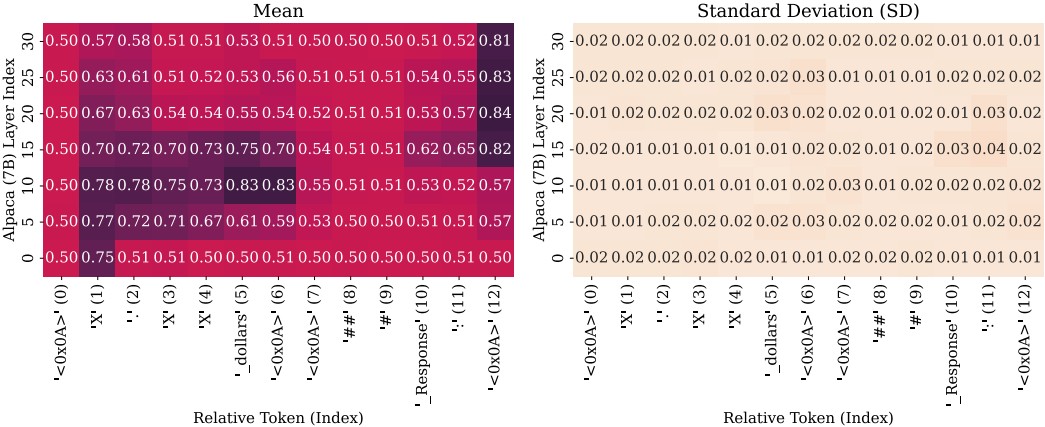

Figure 11: Interchange Intervention Accuracy (IIA) evaluated with **twenty** different irrelevant contexts generated by GPT4-Aug05-2023. Correlation is **0.99** with the "Left and Right Boundary" causal model.

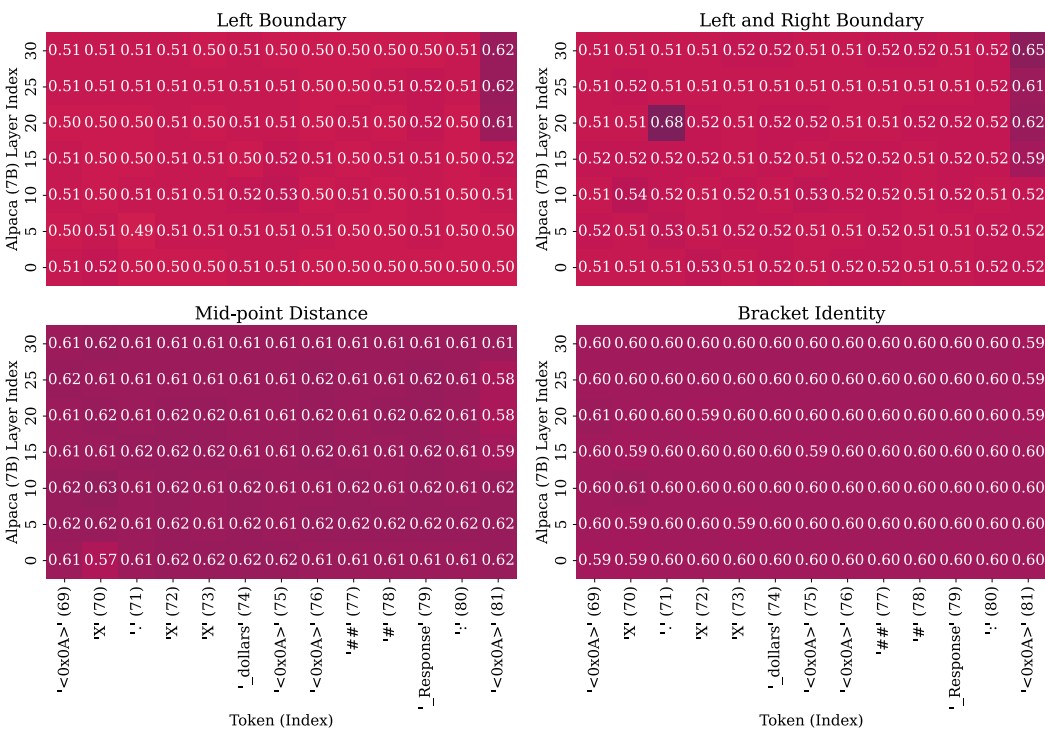

Figure 12: Interchange Intervention Accuracy (IIA) with **random initialized rotation matrix** and learned optimal boundary indices. We run three random seeds.

