# OpenReview forum: "Interpretability at Scale: Identifying Causal Mechanisms in Alpaca"
_NeurIPS.cc/2023/Conference — NeurIPS 2023 poster_

### Official Review · Reviewer_8NXa · 2023-07-06

**Soundness:** 3 good
**Presentation:** 2 fair
**Contribution:** 1 poor
**Rating:** 4
**Confidence:** 5

**Summary:**

This paper introduces an advancement of Distributed Alignment Search (DAS), termed Boundless DAS, where the brute-force search steps are replaced with learned parameters. This enhancement enables efficient exploration of interpretable causal structures in large language models and therefore results in a scalable interpretable model.

**Strengths:**

1) The increasing significance of LLMs necessitates an evaluation of their interpretability. Developing a scalable method to achieve this goal is undeniably crucial and intriguing.
2) Current interpretability methods are inadequate for LLMs due to scalability issues, rendering them impractical. In contrast, the proposed method offers a solution that enables the interpretation of real-world LLMs, filling this crucial gap.

**Weaknesses:**

1) My primary concern lies with the novelty of the paper, which appears to be exceedingly limited. The paper seems to be primarily an extension of DAS and a direct amalgamation of two existing works, namely DAS and neural PDE [58], without addressing any specific challenges or introducing significant innovative elements.
2) The paper heavily relies on the DAS method as its foundation, and it is imperative to provide a comprehensive explanation of DAS, including its limitations, which are currently absent in the paper. A thorough understanding of DAS and its shortcomings is crucial to fully comprehend the context and motivation of the proposed research.



**Questions:**

Please refer to Weaknesses.

**Limitations:**

The authors have discussed the limitations of their work.

---

> ### Author Rebuttal · Authors · 2023-08-10
>
> *Thanks* for your insightful comments and it **leads us to picture more clearly the improvement we have done compared to DAS [Geiger et. al., 2023] and why is that important**. We further clarify our main contributions besides simply proposing a new method in finding alignments. Here, we address all concerns with point-by-point responses.
>
> > **Q1:** "The paper seems to be primarily an extension of DAS without addressing any specific challenges"
>
> **A1:** **Boundless DAS addresses the challenge of scaling causal explanation methods** to the scale of Alpaca (and beyond, assuming one has the compute budget for the work!) **This is significant** because prior methods do not scale to this level, which means that they do not apply to the most relevant present-day models.
>
> Boundless DAS achieves this scalability by removing essentially all aspects of manual search that limited previous methods. The relevant parameters are now learned directly. The resulting optimization problem is straightforward, which we regard as a virtue.
>
> This is also **the first paper (as far as we know) to offer a causally grounded explanation of instruction-following LLMs, with numerous robustness checks.**
>
> > **Q2:** "A thorough understanding of DAS and its shortcomings is crucial"
>
> **A2:** Thanks for the suggestions. Given an additional page in the next revision, **we will provide a detailed background introduction to DAS [Geiger et. al., 2023]**. In addition, we will add a thorough discussion of DAS in the appendix in our next revision. And we will provide a comparison of time complexity between these two methods.

---

> > ### Comment · Reviewer_8NXa · 2023-08-20
> >
> > Thank you for the detailed explanation of the DAS. It is definitely valuable in enhancing comprehension of the proposed approach. Regarding your comment on the novelty, I'm struggling to identify the distinctive technical contribution of the work. It seems to be a straightforward extension of DAS, incorporating an already existing technique (neural PDS [58]). Additionally, as highlighted by fellow reviewers, this extension hasn't been thoroughly assessed across various tasks, with outcomes predominantly confined to Alpaca. Due to these reasons, I would like to keep my score unchanged.

---

### Official Review · Reviewer_AVCz · 2023-07-07

**Soundness:** 3 good
**Presentation:** 3 good
**Contribution:** 4 excellent
**Rating:** 7
**Confidence:** 3

**Summary:**

Finding alignments between variables in a user-hypothesized causal model for how a task could (or should) be performed and representations in neural networks, termed "causal abstraction", is a mechanistic interpretability technique for understanding how neural networks develop predictions at a granular level by using causal interventions.  Prior work has recently proposed the Distributed Alignment Search algorithm for efficiently and robustly finding this alignment, but it still relies on brute-force search over the dimensionality of neural representations. This paper proposes to circumvent the brute force search by *learning* parameters of an orthogonal rotation matrix to maximize the alignment between a neural network representation's linear subspace and one of the variables in the high-level causal model.

They test their method on one rather narrowly scoped task, which is determining if a price is between two numbers, and use the Alpaca (instruction-tuned LLaMA) 7B decoder-only language model. The authors also test whether the found alignments generalize across settings (both input prompts and output token choices for labels), and find that they do.

Edit: I read the authors' rebuttal and the other reviews & discussions. I did not change my score as only one of the weaknesses (writing/presentation) was alleviated. I still think the paper should be accepted though.

**Strengths:**

Originality:
- The application of causal abstraction (or even other variants, like causal mediation analysis/causal tracing) to large instruction-tuned LMs has not been done; this is highly original to the best of my knowledge.

Quality & Clarity:
- The experiments seem sound. The analysis in sections 4.3 - 4.6 is insightful and interesting. The overall findings of the paper are useful to a larger community for understanding math processing in LMs.
- The writing is clear overall.
- Related works section is comprehensive.

Significance:
- Causal analysis of language models at the representation-level is a very important goal with potentially far-reaching impacts for the broader NLP community. Using instruction-tuned models that are in widespread use today, such as Alpaca 7B, bridges the gap between theory and practice and makes this work more applicable to a wider audience.

**Weaknesses:**

- The paper does not read independently (particularly section 3.1) to a reader unfamiliar with prior work. I had to refer extensively to [32] to grasp what the various terms and variables used meant, and to understand what components of the proposed algorithm are novel w.r.t prior work. If there is not space in the main paper, the appendix should be used to thoroughly explain these concepts in a camera-ready version. The math notation could be improved; see "questions" section.
- The method involves pre-registering some possible symbolic models for performing a task; it cannot discover novel symbolic models that are not specified by the user.
- The method is tested on one narrowly-scoped task (determining if a price is between two numbers) and one model. There is no way to "validate" a mechanistic interpretability method beyond principle or axioms as we don't have the ground truth, so some of the findings are more suggestive rather than extremely concrete conclusions. However, this is a common problem in the subfield and I think this paper is convincing overall.

**Questions:**

- On finding strong alignments between Alpaca’s representations and two of the proposed causal models ("left boundary","left and right boundary")— if the method reveals a faithful abstraction, wouldn't that imply that only one abstraction can have strong alignments with model weights, rather than multiple? Can you explain?
- On writing clarification in Section 3.1-- it would be useful to establish early on that "variables" refers to hidden representations in a NN and "algorithm" to a high-level user-specified symbolic model. Relatedly, the terms "input variables" and "target variables" are a bit confusing as you also have input sequences ("base input", "source inputs") and target model outputs, which are different.
-  I’m not sure I would use the term “largest” to describe a 7B model.
- You use the phrase “human-interpretable” to describe the explanations that DAS produces; it would be good to provide a definition or at least some citations of what this concretely means.
- Lines 170-171 and x-axis in Figure 4: these refer to the same thing; it would be good to give a concrete in-line example in the text as you have done for the core instruction.
- Will you open-source your code?

Some other mathematical notation could be improved. For example,
- what do $j$ and $k$ represent? (I'm assuming $k$ is the number of dataset instances and $j$ the instance being indexed, but this is unstated). $k$ appears doubly-defined in Eqn. 3.
- Set notation is mixed between uppercase variables and brackets
- $\mathbf{s}$ and $\mathbf{b}$ are not vectors but sequences of inputs (i.e., tokens), which is a bit nontraditional notation; introduction of $\textbf{Inputs}$ seems as though it should have been introduced earlier.
- both $\mathcal{N}$ and $\mathcal{M}$ represent a neural network model, both "algorithm $\mathcal{A}$" and "model $\mathcal{C}$" (which is never used after defining) represent a high-level causal model, if I understand correctly.
- no definition of $F$, $F^*$, $\Pi$,  $L$
- I did not follow the connection between target variables $X$, $\mathbf{Z}$, and $\mathbf{N}$.
- $y_p$ and $y$ are more traditionally notated as $\hat y$ and $y^*$; and should keep indexing consistent as $\hat y_i$ and $y^*_i$

**Limitations:**

Yes- sufficiently detailed & comprehensive limitations section.

---

> ### Author Rebuttal · Authors · 2023-08-10
>
> *Thanks* for the useful suggestions. **They lead us to define more clearly how to interpret our results, and existing limitations** of Boundless DAS.
>
> > **Q1:** "The paper does not read independently (particularly section 3.1) to a reader unfamiliar with prior work.”
>
> **A1:** We plan to use the extra page to add background on DAS and add a detailed appendix on DAS.
>
> > **Q2:** "The method involves pre-registering some possible symbolic models for performing a task"
>
> **A2:** Yes, the framework verifies whether a causal model can be aligned with representations. Future works may look at how to automatically create hypotheses for Boundless DAS evaluation.
>
> > **Q3:** "The method is tested on one narrowly-scoped task (determining if a price is between two numbers) and one model."
>
> **A3:** Thanks for the comments. While we only work with the Alpaca-7B and a single task, we provide clear evidence that **Boundless DAS has the potential to scale to many other tasks and LLMs**. At the time of this project, the main bottleneck is the model’s ability to solve simple reasoning tasks as described in Appendix A.2..
>
> We agree that the causal mechanism that is uncovered with Boundless DAS may not be the most fine-grained explanation. However, **the found mechanism grounded by our IIA metrics**, can be used to faithfully steer a model's behavior at the inference time and can generalize well as we have tried.
>
> > **Q4:** "if the method reveals a faithful abstraction, wouldn't that imply that only one abstraction can have strong alignments?"
>
> **A4:** **Not necessarily.** In our case, aligning “left and right boundary” entails aligning “left boundary” since the latter is one of the subspaces of the former. This also means our model implements both instead of one. In a case where we can only align the “left boundary” but not both, we would then observe lower IIA for "left and right boundary".
>
> > **Q5:** "terms like"input variables" and "target variables" are confusing”
>
> **A5:** Thanks! Here is a list of terms we will clarify in the next revision.
>
> - *Input variables:* These are essentially the input settings, or counterfactual example pairs that we use to train Boundless DAS.
> - *Target variables:* These are the hidden variables within the neural networks that we try to find the alignments for.
>
> > **Q6:** "I’m not sure I would use the term “largest” to describe a 7B model.”
>
> **A6:** We agree it is not the largest LLMs. **We will revise the sentence** to be clearer. We do, though, think that the method scales to our largest models, provided one has the hardware and compute budget (and access) to run those models!
>
> > **Q7:** "You use the phrase “human-interpretable” to describe the explanations that DAS produces; it would be good to provide a definition or at least some citations”
>
> **A7:** We will do this, thanks! We can draw more extensively on papers we cite: Lipton 2018, Geiger et al. 2020, Feder et al. 2021, and others
>
> > **Q8:** "Lines 170-171 and x-axis in Figure 4: these refer to the same thing; it would be good to give a concrete in-line example in the text as you have done for the core instruction."
>
> **A8:** Thanks! We will provide a concrete example.
>
> > **Q9:** "Will you open-source your code?"
>
> **A9:** Yes!.
>
> > **Q10:** "Some other mathematical notation could be improved. For example, what do  $j$ and $k$ represent?”
>
> **A10:** You are right! We overload the notation of $j$ and $k$ to represent training instances and aligning variables. We will clarify in our methods section.
>
> > **Q11:** "Set notation is mixed between uppercase variables and brackets"
>
> **A11:** Thanks– we will adjust the notation in the next revision.
>
> > **Q12:** "$s$ and $b$ are not vectors but sequences of inputs (i.e., tokens), which is a bit nontraditional notation; introduction of Inputs seems as though it should have been introduced earlier."
>
> **A12:** Correct. We will make it clearer and use different symbols for sampled source and base inputs.
>
> > **Q13:** "Both $N$ and $M$ represent a neural network model, both 'algorithm' and 'model' (which is never used after defining) represent a high-level causal model, if I understand correctly.”
>
> **A13:** Right. $A$ is our high-level causal model as defined in Line 90. We use $A$ to train our Boundless DAS as in Eqn. 2, where the second part is the counterfactual output as if we were intervening on $A$. We will clarify this further.
>
> > **Q14:** "no definition of $F$, $F^*$, $\Pi$, $L$"
>
> **A14:** $F_N$ represents the causal mechanism of causal variable N before the distributed intervention, whereas $F_N^*$ represents the mechanism after the distributed intervention. For instance, $F_{N}(v)$ means calling the “rest” of the forward function (i.e., the causal mechanism of $F_N$) by setting $N$ with activations $v$. We will clarify $F_N$ and $F_N^*$ in our next revision. $\Pi$ can not be removed from Line 99, we will correct that to use only $\tau$ to describe our alignments.
>
> > **Q15:** "I did not follow the connection between target variables $X$, $Z$ and $N$"
>
> **A15:** $X$ is the input, $N$ represents intervened representations in the original basis, $Y$ is the representation after basis change (e.g., via rotation matrix), and $Z$ is a subspace of $Y$ meaning the actual neurons defined by our trainable boundaries we are intervening on the $Y$ space.
>
> > **Q16:** "$Y_p$ and $y$ are more traditionally notated as $\hat{y}$ and $y*$"
>
> **A16:** We use $Y$ to denote a different thing: $Y$ represents our intervening subspace in the neural representation. We will use a different symbol in the next revision to clarify.

---

> > ### Comment · Reviewer_AVCz · 2023-08-15
> > **Response**
> >
> > Thank you for answering my questions and clarifying notational issues. Based on the authors responses, I am reasonably confident the authors will make the writing in Section 3 clearer and provide more adequate background on the DAS method in the updated version, though I strongly urge the authors to make these changes, as it seems I was not the only reviewer who raised these points.
> >
> > I think that the main claim of the rebuttal that Boundless DAS is the "only method in this class of causal explanation methods that can even be applied at the scale of Alpaca or above" is a bit strong; causal tracing/causal mediation analysis is another method that is likely applicable at this scale, so I would be careful making such a claim without empirical support (e.g., runtimes). Additionally, the authors make claims about scaling beyond the Alpaca model used in the paper here, but I think the paper could benefit from more theoretical or empirical justification (or maybe just clearer writing) about why this is the case, such as done in your response to reviewer iGjk.
> >
> > Regarding my first question and the authors' general response "the network may be abstracted by many other causal models as well. Di fferent causal abstractions highlight di fferent explanatory aspects of model behavior.", I feel the paper would benefit from some discussion of how one might interpret or rectify various different causal models that provide different interpretations of model behavior. Without some means of aggregating them and/or resolving conflicts, the practical utility of this form of explanation might be limited (am I understanding this correctly?)
> >
> > I will keep my score as I feel that paper will be of high impact and tackles a difficult and valuable research question (though I defer to other reviewers who are more confident in their assessment of the math).

---

> > > ### Author Response · Authors · 2023-08-16
> > > **Thank you for your comments. We will revise our paper.**
> > >
> > > Thank you for your recent comments. We really appreciate this input. We are eager to make the work as accessible as possible, and this will help a lot in that regard.
> > >
> > > We will update our methods section to clarify further by addressing the feedback from our reviewers. Additionally, the connection with causal mediation analysis or causal tracing is interesting to explore. They are both intervention-based analyses methods and so there may be ways that we can use Boundless DAS to reduce the burden of searching for intervention sites for that method as well.
> > >
> > > Moreover, we'll expand the discussion in our limitations section to emphasize the assumptions behind our methods. We will also suggest potential future works to enhance the reliability of our interpretation results, such as aggregating or consolidating multiple alignment candidates.

---

### Official Review · Reviewer_ExBe · 2023-07-10

**Soundness:** 2 fair
**Presentation:** 2 fair
**Contribution:** 3 good
**Rating:** 5
**Confidence:** 4

**Summary:**

This paper builds on Distributed Alignment Search by introducing a more efficient learned model to search for a given circuit in a model.  They test this method on a specific task: asking the model whether a given numeral falls between two other given numerals. They have a single task, and find an alignment with 2 possible circuits for accomplishing this task and fail to find a good alignment to 2 other handcrafted circuits, demonstrating that this method may be capable of distinguishing between correct and incorrect circuits.

EDITED to raise soundness score.

**Strengths:**

The approach of boundless DAS itself seems to be a worthwhile one, and I believe that it could work.

The approach scales well, up to Alpaca, and therefore could be very useful in modern models.

I'm impressed by the approach of carefully constructing possible circuits to test, and it seems to work surprisingly well in LLM settings.

The observation that the IIA patterns are more structured for the seemingly correct circuits is an interesting one, though I have some reservations about this as a verification method (in questions).

**Weaknesses:**

An obvious criticism of the method itself is that we lose a degree of interpretability by introducing learned parameters into the interpretation method. How many learned parameters does it take before you’ve just introduced another blackbox as your lens? If the problem is that you don’t trust learned parameters to behave interpretable, how can you ensure that you are behaving interpretably? In particular, it doesn’t seem to get away from the complaint that the authors themselves introduce: failing to find an alignment does not mean that the alignment doesn’t exist. This is a mild criticism, and I still consider the method to be potentially useful.

More significantly, the experiments are not convincing. There is only a single task evaluated. They consider only a single model, Alpaca. There are only four possible circuits considered. All of these factors add up to a fairly unconvincing set of experiments which have limited statistical power. Outside of the main experiments, these issues become more apparent: section 4.5 tests generalization from a fixed input to a different fixed input, with only a single sample considered for each case. A test of generalization with inserted context only considers two different possible contexts. Overall, these experiments are too constrained with too few samples to be convincing; while I don't think that this is cherry picked, results that comprise of case study of only a couple of samples could easily lead to publication bias.

Furthermore, for the price task, the effectiveness of the alignment is measured by IIA, but the baseline for IIA is based on an assumption of complete randomness that I do not think is valid. After all, we have a method that has already found an alignment that maximizes the match with the circuit. I would trust this more if you had, e.g., tested it on random weights or weights trained for a completely different structured task that wasn't actually language modeling.

On another note, 4.5 appears to be the method that they should be using as the main experiment. I don't trust results that train the boundary parameters and alignments on the same inputs that they test on, which appears to be the approach taken in 4.3.

The math could be more carefully explained, with more precise definitions for the variables involved and more detailed intuitions for the method. One way to help is to turn Figure 1 into a step-by-step algorithm diagram.

Minor:
- "Boundary" is an overloaded term (used in both the method and the task). You should pick different wording.
- "We find that Alpaca achieves near-perfect task performance because it implements a simple algorithm with interpretable variables." *Because* is a *very* strong claim. Do they actually support the idea that the simple algorithm is the reason for the task performance? Is performance worse for tasks that don’t have simple algorithms? They do not show either piece of evidence.


**Questions:**

Why would baseline IIA be 50%? You are actively aligning a circuit to the model, so shouldn't you be able to find an alignment that performs above 50% even in a randomly weighted model?

What is a target variable $Z_j$?

What if $F_N$?

Please explain the boundary index variable in more detail.

When you restrict $b_j$ to be "a multiple of $b_{j -1}$ with a factor of 2", do you mean that $b_j = 2 b_{j-1}$ or that it could be any multiple as long as it is also even? Why would you use the notation in this way, rather than specify the variable if it is deterministic?

Is it likely that we could have a very high IIA score without having more structured patterns? Rather than validating the IIA score, the observation of sparser patterns seem to strongly correlated with the score by definition, so I don't entirely trust it as a verification of the faithfulness of the alignment.

What is an intermediate boolean value?

How large is the task dataset used to learn the alignment? Is it different from the set used to test IIA for the alignment?

"We sample 100 experiment runs from our experiment pool and create two groups" Do the runs differ only by the input used? Or by some random seed?

In section 4.6, what does it mean to look at learned boundary width? What does a higher width indicate?

Are the training steps (Figure 5 x axis) training steps for learning the alignment, or for the training of the LM? The number of steps is small so I assume it is for the alignment. What do we learn from the learning curve, that we didn't get from the final values?

**Limitations:**

The authors acknowledge that their method is limited by the fact that not finding an alignment is not evidence that there is no alignment to the given circuit, which is very good. However, the limitations of the experiments themselves are not acknowledged.

This method only applies to synthetic tasks where we know the possible algorithms. This isn't explicitly acknowledged.

---

> ### Author Rebuttal · Authors · 2023-08-10
>
> We *greatly appreciate* the reviewer’s in-depth feedback.
>
> > **Q1:** "how can you ensure that Boundless DAS is behaving interpretably? ... failing to find an alignment does not mean that the alignment doesn’t exist."
>
> **A1:** These are **incisive questions that are worthy of broader discussion**. A few thoughts: **It seems like we have to accept that neural models might store information in a *highly distributed fashion*, as *Smolensky and others* anticipated long ago. Our method is a sort of minimal response, as it allows only a limited kind of representation. **The issue of false negatives is important.** With our method, we can at least search vastly more hypotheses than any prior method, reducing the risks here.
>
> > **Q2:** "Overall, these experiments are not convincing."
>
> **A2:** **While we only evaluate with Alpaca-7B and on a single type of task, our method scales to larger open-source LLMs.** Moreover, **the original paper of DAS tried DAS with a set of tasks with different models and scales. Boundless DAS is strictly more scalable and expressive.
>
> **We ran new experiments with 20 random context generated by GPT-4@Aug05-2023** to validate one of our generalization tests with inserted context in Section 4.5. **The overall correlation of the mean IIA with random contexts inserted with our vanilla “left and right boundary” causal model is 0.99.** These results greatly strengthen our claims about alignment generalization.We want to emphasize that it was **unexpected** to find alignments that generalize given the fact that we freeze the rotation matrix and boundary indices at test time.**
>
> > **Q3:** "tested it on random weights or weights trained for a completely different structured task"
>
> **A3:** **Finding alignments on a model that could not perform the task would be uninformative**: if the LM head is randomly initialized, there is little chance of getting the output tokens space correct (e.g., {True, False} or {Yes, No}).
>
> **Our new results suggest good IIAs do not come for free.** **As shown **in Figure 1 of the attached pdf**, all positions drop significantly with random rotation matrix. These results calibrate IIAs in case of unbalanced counterfactual labels (e.g., each control causal model can reach about **0.60** IIA with a random rotation matrix).
>
> > **Q4:** "I don't trust results that train the boundary parameters and alignments on the same inputs that they test on."
>
> **A4:** Alignments and boundary parameters **are tested with unseen counterfactual pairs for all the sections**. Section 4.5 is more extreme: the inputs are not only unseen but with distribution shifts.
>
> > **Q5:** "I think that the math could be more carefully explained", "'Boundary' is an overloaded term"
>
> **A5:** We will provide a clearer description.
>
> > **Q6:** "'We find that Alpaca achieves near-perfect task performance because it implements a simple algorithm with interpretable variables.' *Because* is a very strong claim"
>
> **A6:** We agree that **the sentence should be clarified** to remove any ambiguity. The Boundless DAS analysis is a causal explanation for the model's success, and that is why we used "because". However, we need to clarify that there may be other explanations that highlight different factors in the model's success, as we saw in our analysis of alternative causal models.
>
> > **Q7:** "shouldn't you be able to find an alignment that performs above 50% even in a randomly weighted model?"
>
> **A7:** If a neural model performs at random with respect to a given high-level model which solves the task, we can know a priori that there is not a causal abstraction relation between the neural model and the high-level model.
>
> > **Q8:** "What is a target variable $Z_j$?”
>
> **A8:** The target variables $Z_j$ are sets of variables and the $X_j$ are variables.
>
> > **Q9:** "What is $F_N$?"
>
> **A9:** $F_{N}$ represents the causal mechanism of causal variable N before the distributed intervention, whereas $F_{N}^*$ represents the mechanism after the distributed intervention.
>
> > **Q10:** "Please explain the boundary index variable in more detail."
>
> **A10:** The boundary index variable marks how many neurons the intervention will use at the end. We penalize larger indexes since we want to push the intervention site to be smaller.
>
> > **Q11:** " do you mean that $b_j$ = 2*$b_{j-1}$ or that it could be any multiple as long as it is also even? Why would you use the notation in this way?"
>
> **A11:** We restrict $b_j$ to be "a multiple of $b_{j-1}$ with **a factor of 2 for simplicity**.
>
> > **Q12:** "Is it likely that we could have a very high IIA score without having more structured patterns?"
>
> **A12:** **We observed that but would not make any conclusion regarding high IIA score and structured patterns** in Figure 4.
>
> > **Q13:** "What is an intermediate boolean value?"
>
> **A13:** In our case study, it is a latent boolean value that is abstracted to check whether the input price is above the lower bracket or below the higher bracket.
>
> > **Q14:** "How large is the task dataset used to learn the alignment? Is it different from the set used to test IIA for the alignment?"
>
> **A14:** We random sample 20K counterfactual pairs to train the alignment. **Test is disjoint from train.**
>
> > **Q15:** "'We sample 100 experiment runs from our experiment pool and create two groups' Do the runs differ only by the input used? Or by some random seed?"
>
> **A15:** **These runs are sampled from our experiment pool with different inputs and different random seeds.
>
> > **Q16:** "In section 4.6, what does it mean to look at learned boundary width? What does a higher width indicate?” "Are the training steps (Figure 5 x axis) training steps for learning the alignment?"
>
> **A16:** Boundary width maps to how many neurons are needed in the rotated space to represent the causal variable we are aligning. Higher width meaning we need more neurons to represent the variable. Fig. 5 x-axis represents the training step for alignment.

---

> > ### Comment · Reviewer_ExBe · 2023-08-11
> > **stress tests**
> >
> > Thank you, you've strengthened your results by checking a wider range of context. I think the method is potentially very valuable, and would eventually like to see this work published when it is ready.
> >
> > However, IIA itself doesn't seem to have any calibration for the user and I have no idea what a low value would entail. I believe that to evaluate your method, you need to compare the IIA to settings where you know that it should be low. To really be convinced that the value computed with your evaluation indicates a legitimate aligned circuit, I would want to see:
> > - IIA after searching a random model (rather than comparing against the correct answer, you can compare against the answer that the model gives without intervention, so it's not true that it would be uninformative)
> > - IIA for a known incorrect circuit of similar complexity to the ones found.
> > - IIA on circuits for a task that is not consistently solved, in order to check whether intervening on the circuit actually corrects the model's output. Can you correct the answer by intervening on this circuit for the small number of cases where alpaca is otherwise incorrect? If not, it seems that this circuit is inadequate to explain the model's behavior even on the constrained task given.
> >
> > > A7: If a neural model performs at random with respect to a given high-level model which solves the task, we can know a priori that there is not a causal abstraction relation between the neural model and the high-level model.
> >
> > The problem is that I want to know the actual evaluation, not your a priori judgment of what the real circuit is. It is very easy to trick yourself into thinking that there is a meaningful circuit when there is not, and I don't think that I can accept this paper without knowing, at a minimum, whether you could trick yourself into believing that model had an interpretable circuit when it was in fact random.
> >
> > > A8,9: The target variables ... are sets of variables and the ... are variables. ....  represents the causal mechanism of causal variable N before the distributed intervention, whereas  ...
> >
> > This really doesn't answer my question very well, sorry. I think that your notation needs some rewriting and also if one of these is supposed to be a set of variables and the other is supposed to be a set of sets of variables, you should change the notation generally to reflect the different types. Generally, you need to work on your notation and define it in more detail so that the math is more readable.
> >
> > >  A10: The boundary index variable marks how many neurons the intervention will use at the end. We penalize larger indexes since we want to push the intervention site to be smaller.
> >
> > So you have already chosen how many neurons the intervention actually uses?
> >
> > > A11: We restrict ... to be "a multiple of ... with a factor of 2 for simplicity."
> >
> > This really doesn't clarify anything. My point of confusion was this exact phase that I already read in the paper. What makes it simpler? Can you actually define this variable? Is it learned or is it preselected?
> >
> > > A13: In our case study, it is a latent boolean value that is abstracted to check whether the input price is above the lower bracket or below the higher bracket.
> >
> > If you know the answer, how is it a latent value?
> >
> > Overall, I am not satisfied as to how to assess a value for IIA, or how reliable your method is. I have described several experiments that could be run to calibrate the metric discovered by your method, but as is you have no actual evaluation for the method itself, instead using it straightforwardly on a very limited setting without any confirmation of what any given score indicates. I like the method, and I like the fact that you are trying to solve the problem of finding circuits at large scales in this way, but I would want to see the metric it computes be better calibrated through more experiments.
> >
> > To be clear, these problems exist in normal alignment search. However, your addition of more learned parameters exacerbates the possibility of finding a coincidentally high performing setting and fooling yourself into believing that the coincidence is meaningful. This is why I would like to see you try things like searching for irrelevant circuits.
> >
> > As a last comment, in your rebuttal you refer to whether this generalizes outside of the case of alpaca. I'm actually much less concerned about whether you generalize other language models, which I would accept you can, than whether you generalize to other tasks and circuits. You are advocating for this method on the basis of what is essentially a case study of searching a single set of parameters for behavior on a single very constrained task.

---

> > > ### Comment · Reviewer_ExBe · 2023-08-11
> > > **a possible way of checking a wider range of tasks**
> > >
> > > You could use the original tasks from geiger et al but with in context learning. I'm not sure if this would work. I just know that I don't entirely trust this method without more experiments and some point of comparison for how easy it would be be misled with the IIA metric.

---

> > > ### Author Response · Authors · 2023-08-11
> > > **Evidence that IIA is Well Calibrated**
> > >
> > > We appreciate the critical assessment of IIA, but we believe that **there are already results in the paper where we have low IIA and show that the method is calibrated**.
> > >
> > > According to our randomized alignment baseline, 50% IIA is the floor for performance on the boundary tasks and 60% IIA is the floor for performance on the mid point and bracket identity tasks. IIA is an accuracy metric, and so a gradient of success should not be surprising, and we consider values below the ALPACA model performance of 85% to be a failure to identify the causal variable. Small differences in accuracy can hide crucial aspects of the task!
> > >
> > > In figure 4 "Left Boundary", we can see that many of the locations we attempt to identify a variable are considered there are less than ten locations that we successfully identified the causal variable. **Of the remaining values, many are at random chance and others show some signal, but are below the threshold of strong evidence**.
> > >
> > > There are also low IIA values for two algorithms In figure 4, "midpoint" and "bracket identity" we see no strong evidence of these variables being represented. **All of the IIA values are low enough to be considered failures to confirm the hypotheses and most are at random chance**.
> > >
> > > **The lower IIA values in our main results provide calibration for the user** and show that this method cannot simply optimize to achieve perfect IIA wherever there is a representation that impacts the output.
> > >
> > > When compute is available, we will run a baseline with a randomly initialized model to add another point of calibration. Compute is scarce for us right now, though.

---

> > > > ### Author Response · Authors · 2023-08-11
> > > > **Follow-up on detailed technical questions**
> > > >
> > > > Follow-up on detailed technical questions
> > > >
> > > > **Here are responses to individual questions in the new responses:**
> > > >
> > > > > **Q8,9 (New):** “The target variables ... are sets of variables and the ... are variables. .... represents the causal mechanism of causal variable N before the distributed intervention, whereas …”
> > > >
> > > > **A8,9:** We will update our notations and method section combining suggestions from all the the reviewers.
> > > >
> > > > > **Q10 (New):** So you have already chosen how many neurons the intervention actually uses?
> > > >
> > > > **A10:** We define the neuron population, not the final neuron number that the intervention uses. For instance, we were to run Boundless DAS over a representation with a dimension of 4096, the initial neurons for the intervention is 4096. The boundary learning process will shrink down from that as we always incentivize for smaller boundaries.
> > > >
> > > > > **Q11 (New):** This really doesn't clarify anything. My point of confusion was this exact phase that I already read in the paper. What makes it simpler? Can you actually define this variable? Is it learned or is it preselected?
> > > >
> > > > **A11:**  We restrict $b_j$ to be "a multiple of $b_{j-1}$ with **a factor of 2 for simplicity** in our case study. This restriction basically tells the alignment to start with the same number of swapping neurons for both the left and the right boundary. **It could be any factor.** The number of $b_j$ variables corresponds to the **learned** number of aligning causal variables at the same location.
> > > >
> > > > Now, we will elaborate our boundary learning process a bit more. Boundary index variable marks how many neurons the intervention will use at the end. We penalize larger indexes since we want to push the intervention site to be smaller. By using the boundary index variable, Eqn. 3 will generate a high-pass filter ($Mask_{s}$) between [$b_{j-1}$, $b_{j}$] and the gating factor within the bounds will be approximately 1.0 and approximately 0.0 outside the bound. We then apply this filter to the source representations, and the inverse filter (1.0 - $Mask_{s}$) to base representations before adding them up to get our "soft-intervened" representations. We will revise this section to be clearer in the next revision.
> > > >
> > > > > **Q13 (New):** If you know the answer, how is it a latent value?
> > > >
> > > > **A13:**  Sorry about the confusion. We use “latent” to refer to variables that are not in the input or output explicitly. For instance, the boolean to check whether the input number is lower/higher than a bound is not explicitly in the input.

---

> > > > > ### Comment · Reviewer_ExBe · 2023-08-11
> > > > > **with a factor of 2 for simplicity**
> > > > >
> > > > > Could you find some way to paraphrase this? I just don't have any idea what this phrase is supposed to mean and it's not explained in the paper. So you've explained it to me three times (including in the paper), but each time you have used the exact same phrase, which I still don't understand.

---

> > > > > > ### Author Response · Authors · 2023-08-11
> > > > > > **more clarifications on "a factor of 2 for simplicity"**
> > > > > >
> > > > > > In our experiments, **we align at most 2 high-level variables**, when we learn left and right boundary variables. This means we only need to set two boundary variables to **split up a representation into three parts**:
> > > > > > - (1) the part aligned with the left boundary
> > > > > > - (2) the part aligned with the right boundary
> > > > > > - (3) the part not aligned with any causal variable.
> > > > > >
> > > > > > **For convenience, we assumed that the dimensionality of the first two parts was equal** (i.e. the boundaries between variables related by $b_2$ = 2*$b_1$). This could be relaxed but seemed reasonable for the experiment in which the two variables were simply the left and right bracket values.
> > > > > >
> > > > > > In general, we can learn any number of boundaries, and we could alter the relationship between boundaries to be any differentiable function or independent.
> > > > > >
> > > > > > You may refer to our code for a better understanding. You can find this in **L245** in `code/models/llama/modelings_alignable_llama.py` in our attached `.zip` file.

---

> > > > > > > ### Comment · Reviewer_ExBe · 2023-08-14
> > > > > > > **factor of 2**
> > > > > > >
> > > > > > > So what you mean here is that you are restricted to a *branching* factor of 2 in the circuit?

---

> ### Comment · Reviewer_ExBe · 2023-08-14
> **general comment**
>
> I think that the approach here is potentially very valuable, but this paper remain a case study of a single task, and the evidence is extremely limited for the reasons I've given. It provides a single very limited example without any further validation of the method, e.g., by showing that correcting the circuit found can fix an incorrect judgment.

---

> > ### Author Response · Authors · 2023-08-14
> > **New Baseline with Randomly Initialized LLM**
> >
> > Thank you for your continued engagement. We were able to pursue a version of the random LM experiment you suggested, and this proved fruitful. We randomly initialized a model, which resulted in near 0% task performance on the price checking task, as expected. After running boundless DAS on first layer of the random LLM, the alignments to each token's residual stream ranged from 0 % to 69% IIA,  which is comparable to a most frequent label dummy model (66%). For the representations with near chance IIA, this means that we were able to find a distributed neural representation that shifts the probability mass to "yes" or "no", but does not differentiate between the two. We also found that we could find a distributed neural representation that shifts the probability mass to the tokens "dog" and "give" instead of "yes" and "no", so this really is random causal structure that emerges from the massive number of parameters in the randomly initialized LLM. It's quite a neat result we plan to include in the paper!
> >
> > **Crucially**, we found that IIA drops to 0 for this run on all of our robustness checks from the paper. In addition, it drops to 0 if we use it on a different random LM. The results really nicely complement the ones we reported earlier for random rotation matrices on the highly structured, pretrained Alpaca model. These results will help calibrate people on the metric!

---

> > > ### Comment · Reviewer_ExBe · 2023-08-14
> > > **random baseline**
> > >
> > > Thank you, that finding clarifies things and also supports my point that the robustness check should be the *default* way of testing a circuit, rather than being considered a stress test. Would you be willing to slightly restructure in that way?
> > >
> > > I still would rather see more tasks, but the results from the random baseline satisfy me that at least the robustness test isn't completely broken as a validation method (although it appears that checking IIA without the distribution shift is somewhat broken, and outperforming a random baseline isn't completely satisfactory as a validation of the metric).
> > >
> > > Currently, while I would rather see more tasks rather than a case study on a single task, I'm willing to raise my score slightly reflecting the useful validation you've done. Don't be discouraged that it's still not a high score, as I think that the method is potentially very valuable.

---

> > > > ### Author Response · Authors · 2023-08-14
> > > >
> > > > We agree that these robustness tests are crucial evidence for our conclusions, and we will restructure to emphasize that our robustness checks and random baselines are crucial evidence for our conclusions and that they should be the default. Thank you for the meaningful critiques that lead to informative experiments!

---

### Official Review · Reviewer_SQSS · 2023-07-19

**Soundness:** 3 good
**Presentation:** 3 good
**Contribution:** 2 fair
**Rating:** 6
**Confidence:** 2

**Summary:**

This work proposes an extension to Distributed Alignment Search (DAS) called "Boundless DAS" that replace the brute-force search with learned parameters in order to scale the method to large language models (LLMs).
The method evaluated on the Alpaca 7B model tasked to output "yes" or "no" to a simple numerical problem such as "is X between Y and Z?".
Experimental analysis showed that the method is able to scale and find an alignment with two potential causal mechanisms.
Further experiments also show that the method is robust to slight modifications in the prompt of the model (additional sentence in the input, change of output format).

**Strengths:**

This work proposes a new explainability method that is scalable to LLMs, which is an important research question as the field quickly progresses towards larger models that become part of real-life products. Hence, research to understand the inner mechanisms of LLMs is very important.

The proposed method is well-tested on various input/output perturbations of the target task, showing the robustness of Boundless DAS.

**Weaknesses:**

- the technical aspect of the paper is challenging to understand for someone that is not familiar with the field of causality

- the scope of the task used to evaluate the method is very narrow. Some discussion on how this can be applied to more critical scenarios where interpretability is required (such as the medical, financial, or legal domains) could improve the paper.

- It is mentioned that a good causal model could also explain the errors of the system. However, the current method does not explain why the model fails in certain examples.

**Questions:**

- Out of four possible causal mechanisms, two seem more plausible, but it is not clear which one is the one used by the model: "left boundary" or "left and right boundary"? Can we identify which causal method is the most likely being used, or is it the case that the model can use one or the other depending on the input?

**Limitations:**

Some limitations are mentioned.
However, one limitation of this work that should be mentioned is the relatively narrow scope of the target task being evaluated. This may be ok if the authors discuss how to transfer their method to more "realistic" scenarios in which interpretability is required (such as the medical, legal, or financial fields). Another possibility is that it is challenging or impractical to apply the same type of analysis in more complex scenarios. Either way, the paper could benefit from this type of discussion.

---

> ### Author Rebuttal · Authors · 2023-08-10
>
> We *thank* the reviewer's *thoughtful comments*. We **address the major concern around the applicability of Boundless DAS as well as how to interpret alignment results** where multiple causal models reach good interchange intervention accuracy (IIA). Here, we address all concerns with point-by-point responses.
>
> > **Q1:** "the technical aspect of the paper is challenging to understand for someone that is not familiar with the field of causality"
>
> **A1:** Thanks for your comment! Given the page limit, we did not include a detailed introduction to DAS [Geiger et. al., 2023]. **In the next revision, we will provide a full detailed discussion of DAS** in the appendix in the next revision to ground Boundless DAS. We will **sharpen the notation in Section 3**. We also want to note that we provide a full pseudo-code implementation of Boundless DAS in Appendix A.5; this may provide additional clarifications.
>
> > **Q2:** "the scope of the task used to evaluate the method is very narrow."
>
> **A2:** While it is true that we only analyze the Alpaca model with a very specific task, **the goal of this paper is to present an alignment search method that works well with LLMs such as Alpaca-7B and beyond**. We see no obstacle to applying the methods for additional models and tasks. We will release our codebase that scales with any decoder-only model. The main bottleneck for us to try other tasks with Alpaca is that it is hard to find reasoning tasks that Alpaca does well. We outlined some of our task finding procedures in Appendix A.2. For domain experts working with stronger models, this kind of exploration might be easier.
>
> > **Q3:** "the current method does not explain why the model fails in certain examples."
>
> **A3:** **This is a good point and suggests great next steps!** At present, errors could be explored by seeing how their computations do not conform to the high-level model under investigation. One could also hypothesize high-level causal models of the errors and assess them as explanations for the underlying problem.
>
> > **Q4:** "Out of four possible causal mechanisms, two seem more plausible, but it is not clear which one is the one used by the model: "left boundary" or "left and right boundary"?"
>
> **A4:** In this case, aligning “left and right boundary” actually **entails** aligning “left boundary” since the latter is one of the subspace (i.e., the subspace for the left boundary) of the previous causal model. In other words, **the “left boundary” causal model can be seen as an abstraction of the “left and right boundary” causal model** but not the other way around. This also means our model implements both checks instead of one. In a case where we can only align the “left boundary” but not both, we would then observe lower IIA for "left and right boundary".

---

> > ### Comment · Reviewer_SQSS · 2023-08-11
> > **response to authors**
> >
> > Thank you very much for your clarifications, the contributions are now more clear to me.
> >
> > I would still like the authors to discuss in their revised paper the limiting factor for this method, which is (if I understand correctly according to "_The main bottleneck for us to try other tasks with Alpaca is that it is hard to find reasoning tasks that Alpaca does well._") that in order to find a causal alignment, the model must first be very good at solving the task on its own.
> >
> > I think this is very important to highlight for transparency reasons in the revised paper. The revised paper will benefit from some discussion around this, like why is this the case, and what future work could help mitigate this limitation.
> >
> > Thank you for your time and effort in this important research direction.

---

> > > ### Author Response · Authors · 2023-08-11
> > > **Thank you for your comments. We will revise our paper.**
> > >
> > > Yes, we will add discussion of this – thanks! It's really primarily an issue in the phase where we are seeking to motivate Boundless DAS. For that, we need to try to be sure that there is a systematic solution to identify. If we accept that Boundless DAS is a useful tool, then we can apply it to an under-performing model. It could be used to show that the model conforms to a very strange and problematic high-level causal model, or to accumulate some indirect evidence that the model doesn't implement any reasonable causal models. But one does need to trust the method at this stage, whereas we are currently seeking to build an argument for the method itself.

---

> > > > ### Comment · Reviewer_SQSS · 2023-08-11
> > > > **got it**
> > > >
> > > > oh I see what you mean! thank you.
> > > >
> > > > At this stage, I would like to see more verifications like yours on additional tasks before fully accepting Boundless DAS and using it on underperforming models, but your work is definitely a good first step in that direction. Thanks.

---

### Official Review · Reviewer_6cS3 · 2023-07-20

**Soundness:** 4 excellent
**Presentation:** 4 excellent
**Contribution:** 4 excellent
**Rating:** 8
**Confidence:** 3

**Summary:**

This paper introduces a method called Boundless DAS, an algorithm that can automatically test the "alignment" between human-specified symbolic algorithms and the computational structure found in the weights of a neural network. The authors apply Boundless DAS to a real-world 7B parameter language model called Alpaca and find evidence of the implementation of a particular symbolic algorithm that computes whether some number $q$ is in the range $[a, b]$.

**Strengths:**

Really cool paper!
* I like the idea of using gradient descent to create a correspondence between a symbolic algorithm and a set of neural network features. Continuous optimization feels a little easier to tackle (initially) than discrete combinatorial stuff.
* It's impressive that this works on a real-world, billion-parameter model.
* While the tricks to go from DAS to Boundless DAS seem reasonable and slick, my favorite part is the empirical section — inspecting the alignments that the algorithm discovers, looking for generalization, etc. Well done.

**Weaknesses:**

Not much here. If I had to say anything, the paper is quite dense and hard to follow. For newcomers, it requires somewhat extensive background reading on causal abstraction and the previous DAS work(s). Otherwise, the writing is fairly clear, ideas are well-motivated, claims are well-supported and analyzed, etc.

**Questions:**

* What kinds of tasks have you tried DAS or Boundless DAS on? Do you ever find gradient descent struggling to find a faithful solution? Do you have intuitions for what kinds of tasks this works well on, and what it doesn't? How about algorithms with lots of input variables, or complex ones requiring lots of intermediate ones?
* I'm trying to get a sense for how far we can push Boundless DAS; what do you think? For any challenges you foresee, what do you think the right next steps are?

---

> ### Author Rebuttal · Authors · 2023-08-10
>
> We *thank* the reviewer's *inspiring comments about future works* that can be built on top of Boundless DAS. We **provide more discussions about possible future extensions**, and we will revise our paper to reflect these additional thoughts. Here, we address all concerns with point-by-point responses.
>
> > **Q1:** "it requires somewhat extensive background reading on causal abstraction and the previous DAS work(s)."
>
> **A1:** Thanks for the comment! We plan to **use the additional page to add an introduction to DAS** [Geiger et. al., 2023], and add **a full detailed appendix on DAS** to provide readers more context about the background, and how Boundless DAS improves upon it.
>
> > **Q2:** "What kinds of tasks have you tried DAS or Boundless DAS on? Do you ever find gradient descent struggling to find a faithful solution? Do you have intuitions for what kinds of tasks this works well on, and what it doesn't? How about algorithms with lots of input variables, or complex ones requiring lots of intermediate ones?"
>
> **A2:** **The original DAS paper [Geiger et. al., 2023] provides a set of examples including synthetic logic inference task as well natural language inference (NLI) task with smaller models.** Earlier causal abstraction ideas have been applied to other NLI problems, ground language understanding tasks, and structured image tasks. We haven’t tried Boundless DAS on most of these, but Boundless DAS is a scalable and more general approach, so such applications would yield as good or better results.
>
> **For the current work, we wanted to focus on tasks that LLMs solve well with no fine-tuning.** The bottleneck for us to try other tasks is, given our compute and time limit, we struggled to find a publicly-available instruction-following model that solved interesting  reasoning tasks in a zero-shot fashion (Appendix A.2). We did find failure cases such as those control conditions mentioned in the paper! These control models are interpretable models but we fail to find clear evidence of alignments. It has so far not been the case that gradient descent gets stuck in local optima, though this is certainly possible.
>
> **On aligning more complex causal graphs: Yes! We are working on it right now.** In this paper, we are only aligning individual causal variables, or a high-level model with a single layer of abstraction. The next step is to align a full, multilayer causal graph with a neural model, and evaluate the faithfulness of the alignment more rigorously.
>
> > **Q3:** "I'm trying to get a sense for how far we can push Boundless DAS, …, what do you think the right next steps are?"
>
> **A3:** **The biggest obstacle we see is the need to hypothesize a high-level causal model.** All explainability methods require a hypothesis of some sort, but often the burden seems smaller and so is less of a blocker to people running the analysis (e.g., for probing, one picks a supervised learning task; for feature attribution, one picks a set of neurons to analyze). We would like to  automate the process of proposing high-level models combined with Boundless DAS to generate a faithfulness hypothesis or circuit that is realized by the model.
>
> Another interesting challenge would be **applying mechanistic interpretability tools to controlled language generation.** Thus far, we have only applied Boundless DAS to a single token or fixed length language generation tasks. It would be interesting to see how a token-level intervention through Boundless DAS causally affect the iterative decoding process where the generation length is unbounded.
>
> On the application side, a good counterfactual alignment (even though it is not a structural alignment) helps us steer the model in a robust way. One future step we are thinking to take is to align, for instance, story genre variables in the LLMs and **steer the model at the inference time** to generate different genres given a prompt starter.

---

> > ### Comment · Reviewer_6cS3 · 2023-08-19
> >
> > Thanks for the careful and detailed replies! I appreciate the inclusion of further DAS background. I think this is an interesting & potentially-valuable direction that deserves to be investigated further and discussed in the community.

---

### Official Review · Reviewer_iGjk · 2023-07-27

**Soundness:** 3 good
**Presentation:** 1 poor
**Contribution:** 3 good
**Rating:** 6
**Confidence:** 2

**Summary:**

This paper proposes a method for interpreting neural models by learning an alignment of their representations to a specified interpretable causal model. The method extends prior work, namely Distributed Alignment Search (DAS), by addressing a key limitation that prevents the scaling of DAS. The limitation is overcome by learning a set of masks that selects subsets of neural representations to align to different parts of the interpretable causal model. The method is evaluated experimentally by aligning Alpaca representations to a set of correct and incorrect hand-written causal models.

**Strengths:**

* The problem of interpreting large models is important.
* The approach is reasonable and scales a previous approach.
* The experimental results were convincing.

**Weaknesses:**

* Section 3 was hard to read. See the questions below.
* Section 3 did not feel self-contained. The source of intractability in DAS did not jump out to me.
* Giving the computational complexity for DAS and boundless DAS would make the efficiency argument stronger, unless they are the same.



**Questions:**

1. What is the value of introducing variable $Z$? It looks like only $X$ are used after section 3.1
2. Is $\Pi$ on line 99 used anywhere else?
3. In line 115, '$b_j$ is restricted to be a multiple of $b_{j-1}$ with a factor of 2'. However, $b_0 = 0$. What does this mean?
4. Why are target variables $Z_j$ bold but high-level variables $X_j$ not?
5. In Eq. (1), what is the difference between $F_N^*$ and $F_N$? Is $F_N$ defined?
6. What is the problem that was brute-forced in DAS, that $M$ learns to solve? Section 3.1 would be the best place to highlight intractability.
7. Line 37 would be another nice place to highlight limitations in scaling DAS. What is the largest dimensional representation it has been applied to?
1. What is the runtime improvement due to the method?
8. Line 216: "expectation"

Clarifying the shortcoming of DAS, section 3, and giving the computational complexity would convince me.

**Limitations:**

The limitations were adequately addressed.

---

> ### Author Rebuttal · Authors · 2023-08-10
>
> We *thank* the reviewer's *great suggestions on detailing improvements* over DAS and **highlights the scalability that our method brings**. We will use the additional page to clarify our improvements and provide a detailed introduction to the DAS method [Geiger et. al., 2023]. Here, we address all concerns with point-by-point responses.
>
> > **Q1:** "Section 3 did not feel self-contained, ..., Giving the computational complexity for DAS and boundless DAS would make the efficiency argument stronger."
>
> **A1:** We will revise our draft to make the improvements to DAS clearer, focusing on how the changes enable us to scale. In short, DAS learns a rotation matrix but requires manual search to determine how many neurons are needed to represent the aligning causal variable. Boundless DAS automatically learns boundaries (i.e., how many neurons are needed is determined by the “soft” boundary index via a boundary mask learning). For instance, given a representation with a dimension of 1024, DAS should in principle be run for all lengths k from 1 to 1024. In practice, this would be infeasible, so some subset of the lengths need to be chosen heuristically, which risks missing genuine structure. For Boundless DAS, we turn this search process into a mask learning process. See our comments below for more detailed comparison in terms of time complexity.
>
> > **Q2:** "What is the value of introducing the variable $Z$? It looks like only $X$ are used after section 3.1"
>
> **A2:** The variable $Z$ is only used to define the interchange intervention operator and the projection operator, and plays no "contentful" role in the definitions. $X$ is the original representation space. We will make this clearer!
>
> > **Q3:** "Is $\Pi$ on line 99 used anywhere else?"
>
> **A3:** Thanks for catching this notational error. It should be just $\tau$ representing the alignment mapping between a high-level variable and neural representations.
>
> > **Q4:** "In line 115, $b_j$ is restricted to be a multiple of $b_{j-1}$ with a factor of 2'. However,. What $b_0$ does this mean?"
>
> **A4:** Since b_j marks the boundaries here, b_0 is equal to 0 as the starting index of the boundary. Essentially, Eqn. 3 will generate a high-pass filter ($Mask_{s}$) between [$b_{j-1}$, $b_{j}$] and the gating factor within the bounds will be approximately 1.0 and approximately 0.0 outside the bound. We then apply this filter to the source representations, and the inverse filter (1.0 - $Mask_{s}$) to base representations before adding them up to get our "soft-intervened" representations. We will revise this section to be clearer in the next revision.
>
> > **Q5:** "Why are target variables $Z_j$ bold but high-level variables not $X_j$?"
>
> **A5:** The target variables $Z_j$ are sets of variables and the $X_j$ are variables. We wanted to indicate this type difference with the bold typeset.
>
> > **Q6:** "In Eq. (1), what is the difference between $F_{N}^{*}$ and $F_N$ ? Is $F_N$ defined?"
>
> **A6:** $F_{N}$ represents the causal mechanism of causal variable N before the distributed intervention, whereas $F_{N}^*$ represents the mechanism after the distributed intervention. For instance, $F_{N}(v)$ means calling the “rest” of the forward function (i.e., the causal mechanism of $F_N$) by setting $N$ with activations $v$. We will clarify $F_N$ and $F_{N}^{*}$ in our next revision.
>
> > **Q7:** "What is the problem that was brute-forced in DAS, that M learns to solve? Section 3.1 would be the best place to highlight intractability."
>
> **A7:** Thanks again for raising this issue. This helps us to better ground our work and highlights our improvements. As we clarified above, Boundless DAS is approximately **O(N*m)** quicker where N is the number of population dimensions and m is the number of causal variables we are aligning. In the case of LLMs, it scales DAS to models with billions of parameters like Alpaca-7B and beyond.
>
> > **Q8:** "Line 37 would be another nice place to highlight limitations in scaling DAS. What is the largest dimensional representation it has been applied to?"
>
> **A8:** The largest dimension is BERT with a hidden dimension of 768 with two variables in DAS [Geiger et. al., 2023]. And the result in its original paper is also not guaranteed to be optimal since only a limited combination of dimension settings are tried. In this paper, each aligning representation is with a dimension of 4096 with two variables that are aligned simultaneously.
>
> > **Q9:** "What is the runtime improvement due to the method?"
>
> **A9:** It is **O(N*m)** quicker where N is the number of total dimensions and m is the number of causal variables we are aligning. Essentially, to brute-force search over different dimensionality for intervention on a token representation with a size of 4096, we need to scan through {1,2,3,...,4096} each covering a single data point for a single aligning causal variable depicting how many neurons we are swapping values for. With Boundless DAS, we only need to run it once.
>
> > **Q10:** "Line 216: 'expectation'"
> **A10:** Thanks! We will correct the typo in the next revision.
>
> > **Q11:** "Clarifying the shortcoming of DAS, section 3, and giving the computational complexity would convince me."
>
> **A11:** Thanks for the great suggestions. We believe this concern is addressed in responses above! We will add the discussion around complexity improvements a bit in the next revision.

---

> > ### Comment · Reviewer_iGjk · 2023-08-17
> >
> > Could you also provide the full computational complexity of DAS? I believe my concerns are addressed in the rebuttal. As all the concerns were presentation-related, I can only increase my score to weak accept without seeing the next version.

---

### Author Rebuttal · Authors · 2023-08-10

We thank the reviewers for their incisive comments and questions. In this comment, we summarize our overall response to major points, emphasizing new experiments. This is followed by point-by-point responses.

1. **On the question of novelty and utility**, we note that Boundless DAS is the only method in this class of causal explanation methods that can even be applied at the scale of Alpaca or above. This is a phase change in terms of what is possible -- after all, models this size and larger are the most relevant to the field right now.

1. We also want to take this opportunity to clarify **what Boundless DAS or DAS [Geiger et. al., 2023] could explain.** Having a good alignment means that the high-level model is a causal abstraction of the neural network, in terms of both factual and counterfactual behaviors. The two models can have different structure, and the network may be abstracted by many other causal models as well. Different causal abstractions highlight different explanatory aspects of model behavior.

1. **How to interpret the results when both causal models "left boundary" and the "left and right boundary" find strong evidence in Section 4.3?** The second model strictly extends the first, but the network may be implementing both solutions. The strong IIA scores show that one can reason about either high-level model and not be misled. We elaborate this point in our responses to *Reviewer SQSS* and *Reviewer AVCz*.

1. **Reviewers note that we report experiments only for Alpaca.** We do try to broaden this in Appendix A.2 to other models in this class, but we find publicly accessible models often fail to solve these reasoning tasks robustly. We argue that our experiments achieve our primary goal of showing that causal explanation methods are effective at this scale. We have a limited compute budget (and limited space in the paper), but we've shown that Boundless DAS can easily scale beyond Alpaca when budget allows. We will also release our code that is compatible with any decoder-only model.

1. **Building on the above: the original DAS paper [Geiger et. al., 2023] reports on a number of other experiments, and prior work on causal abstraction includes even more diverse experiments.** Boundless DAS is a more scalable and general version of all these methods, and so we think all this accumulated evidence ultimately supports the applicability of Boundless DAS for causal explanations across many models and tasks.

1. **New experiments are added to validate our main results in Section 4.3.** The reviewers asked for more baselines to more fully contextualize our results. To this end, we report IIAs with the same settings with random rotation matrices in the attached pdf. (We are not able to explore randomly initialized LMs because IIA is ill-defined if the model is not able to perform the task.) As shown in **Figure 1 of the attached pdf**, IIA score drops from **0.83** to **0.53** at layer 10, token position 75 for our “left and right boundary” causal model. Other positions drop significantly as well. These results also help us to calibrate IIAs in case of unbalanced labels (e.g., our two control causal models reach **0.60** and **0.61** IIA at the same location with a random rotation matrix). These results suggest that good IIAs for our first two causal models do not come for free. We will include these baseline IIAs in the next revision to better contextualize our findings.

1. **New experiments are added to validate our alignment generalization ability in Section 4.5.** Reviewer ExBe asked about tightening up the series of experiment-based arguments we offer in Section 4.5. To this end, we greatly expanded the range of inputs we consider, using **20 random contexts generated by GPT-4@Aug05-2023** with our “left and right boundary” causal model. Mean IIAs along with prompts we used are included in the attached pdf. As shown in **Figure 2 of the attached pdf**, the averaged IIA at layer 10, 6-th token position (relative since we have prefixes at different lengths) is **0.83** with a standard deviation of 0.02. This aligns with two prompts we reported in the paper (**0.80** for the first, and **0.84** for the second prompt as shown in Figure 9 of the Appendix). The overall correlation of IIAs with our vanilla “left and right boundary” causal model is **0.99**. These results *greatly strengthen* our claims about alignment generalization. We will update current results with these new prompts in the next revision.

1. **We focus on success cases for Alpaca, but we could use the method to understand errors.** This would proceed in the same mode as in the paper: one would hypothesize a source of the errors and use Boundless DAS to assess the idea. Reviewer SQSS suggested this and we are eager to explore it in future work.

1. **On writings and notations clarifications and suggestions:** The reviewers offered numerous suggestions for **improving terminology and notation**, and improving the paper overall. We are grateful for all this input and will make use of it in our next revision.

---

### Decision · Program_Chairs · 2023-09-21

**Decision:**

Accept (poster)

**Comment:**

Meta Review for Interpretability at Scale: Identifying Causal Mechanisms in Alpaca

As reviewer iGjk wrote, this work proposes a method for interpreting neural models by learning an alignment of their representations to a specified interpretable causal model. The method extends prior work, namely Distributed Alignment Search (DAS), by addressing a key limitation that prevents the scaling of DAS. The limitation is overcome by learning a set of masks that selects subsets of neural representations to align to different parts of the interpretable causal model. The method is evaluated experimentally by aligning Alpaca representations to a set of correct and incorrect hand-written causal models.

Many reviewers commented on the strength of this work, as reviewer SQSS wrote, presents a explainability method that is scalable to LLMs, which is an important research question as the field quickly progresses towards larger models that become part of real-life products. Hence, research to understand the inner mechanisms of LLMs is very important.

There are no clear very negative issues or weaknesses. The reviewer's feedback has been incorporated into the manuscript, and I believe it will be a paper that will offer good value and interest to the NeurIPS community.